# OPTIMAL REPRESENTATIONS FOR COVARIATE SHIFT

**Yangjun Ruan**[*12]**, Yann Dubois**[*2]**, Chris J. Maddison**[12]
[1]University of Toronto & [2]Vector Institute
`{yjruan,yanndubois,cmaddis}@cs.toronto.edu`

## ABSTRACT

Machine learning systems often experience a distribution shift between training and testing. In this paper, we introduce a simple variational objective whose optima are *exactly* the set of *all* representations on which risk minimizers are guaranteed to be robust to any distribution shift that preserves the Bayes predictor, e.g., covariate shifts. Our objective has two components. First, a representation must remain discriminative for the task, i.e., some predictor must be able to simultaneously minimize the source and target risk. Second, the representation's marginal support needs to be the same across source and target. We make this practical by designing self-supervised objectives that only use unlabelled data and augmentations to train robust representations. Our objectives give insights into the robustness of CLIP, and further improve CLIP's representations to achieve SOTA results on DomainBed.

## 1 INTRODUCTION

It is hard to build machine learning (ML) systems that are robust to distribution shifts between a source (train) and target (test) domain. One promising approach to domain generalization (DG) is learning robust representations from which predictors trained on source must perform well on target. In practice, however, no current DG methods for learning representation uniformly outperform empirical source-risk minimizers (ERM) (Gulrajani & Lopez-Paz, 2021). Furthermore, our theoretical understanding of DG is still lacking. Specifically, while previous work have studied properties that would or would not imply robust representations (Ben-David et al., 2007; 2010a; Zhao et al., 2019; Johansson et al., 2019), the *minimal* set of *achievable* requirements for perfect DG is not yet known.

We introduce the first, simple, variational objective whose optima are exactly the set of all representations on which source risk minimizers are guaranteed to generalize across distribution shifts that preserve the Bayes predictor. We work in an idealized DG (IDG) setting; we assume that a learner has access to the source population risk. Our variational characterization implies that it is both sufficient and *necessary* for optimal IDG that a representation: (a) remains discriminative for the learning task, i.e., there must exist predictors from the representation to the labels that can simultaneously minimize *both* source and target risk; and (b) keeps the support of its marginal distribution invariant to shifts.

This means that any optimal representation learning method must seek discriminative information about the target. Even worse, we prove that without access to some knowledge about the target, any representation learning algorithm cannot uniformly (over all target domains) outperform a *constant* representation, which may explain why DG methods struggle to outperform ERM.

We show, in theory and practice, how to overcome these challenges using only a large set of unlabeled examples and particular data augmentations that retain all discriminative information but minimal domain-specific information. Text descriptions of images are examples of such augmentations, as they are informative for many downstream classification tasks, but they remove a lot of domain-specific information. With such augmentations, we design practical self-supervised learning (SSL) objectives for learning robust representations. Our objectives give insights into the robustness of CLIP (Radford et al., 2021) over other SSL methods, and lead to improved CLIP-based representations that achieve state-of-the-art (SOTA) results on DomainBed (Gulrajani & Lopez-Paz, 2021). To summarize, we:

- provide minimal sufficient objectives whose optima achieve optimal DG under covariate shift;
- prove that it is impossible to learn useful representations without accessing target information;
- provide practical objectives to learn optimally robust representations using specific augmentations;
- obtain SOTA results on typical domain generalization benchmarks.[1]

---

[*]Authors contributed equally.
[1]Our implementation is released at `https://github.com/ryoungj/optdom`.

## 2 BACKGROUND: DOMAIN GENERALIZATION AND REPRESENTATIONS

We are interested in predictions that are robust across distribution shifts. We formalize this using domain generalization (DG) language. Given a distribution $p_{X,Y \mid d_s}$ over inputs $x \in \mathcal{X}$ and labels $y \in \mathcal{Y}$ from the *source domain* $d_s \in \mathcal{D}$, we select a predictor $f : \mathcal{X} \to \Gamma$. The predictions $\gamma \in \Gamma$ could for example be labels or distributions over labels. Despite being selected on the source domain, we would like $f$ to achieve a small expected risk with respect to a loss function $\ell : \mathcal{Y} \times \Gamma \to \mathbb{R}_{\geq 0}$,

$$\mathrm{R}_f^d [Y \mid X] := \mathbb{E}_{p_{X,Y \mid d}}[\ell(Y, f(X))], \tag{1}$$

on a distribution $p_{X,Y \mid d}$ from a *target domain* $d = d_t \in \mathcal{D}$, which is somehow related to $d_s$.

A common strategy for DG is to learn robust representations, which splits the problem into two. First, learn an encoder $p_{Z \mid X}$, which maps inputs $X$ to representations $Z$. Then, learn a predictor $h : \mathcal{Z} \to \Gamma$ from representations $Z$ to labels $Y$ using standard risk minimization. The goal is to design a robust representation $Z$, so that predictors $h$ trained to minimize the source risk $\mathrm{R}_h^{d_s} [Y \mid Z]$ also achieve low target risk $\mathrm{R}_h^{d_t} [Y \mid Z]$. Many methods have been proposed to try to learn such $Z$, e.g., by enforcing domain invariance of the marginal $p_{Z \mid d}$ (e.g., Ganin et al., 2016). Still, many of these proposals are not sound (Zhao et al., 2019; Johansson et al., 2019). Furthermore, they rarely outperform source empirical risk minimization (ERM) in practice (Gulrajani & Lopez-Paz, 2021).

## 3 OPTIMAL REPRESENTATIONS FOR DOMAIN GENERALIZATION

To separate domain generalization from finite sample generalization, we consider an idealized DG (IDG), where the predictor $h$ is selected on the source *population* risk rather than empirical risk. We assume sample spaces $\mathcal{X}, \mathcal{Z}, \mathcal{Y}, \mathcal{D}$ are discrete; formal statements and proofs are in Appxs. A and B.

### 3.1 DEFINING OPTIMAL REPRESENTATIONS FOR IDEALIZED DOMAIN GENERALIZATION

We want to evaluate the quality of a representation $Z$ of $X$. In our IDG, the learner is given a random source $D_s$; she selects any source risk minimizer; and is scored according to her risk on a random target domain $D_t$. To give uniform guarantees while reflecting the uncertainty over the source-target pair $(D_s, D_t)$, we measure the quality of $Z$ as the expected risk of the learner's worst-case choice.

**Definition.** The *idealized domain generalization risk (IDG risk)* of an encoder $p_{Z \mid X}$ is the expected (over domains) worst-case (over source risk minimizers) target risk, i.e.,

$$\mathrm{R}_{\mathrm{IDG}} [Y \mid Z] := \mathbb{E}_{p_{D_s, D_t}} \left[ \sup_{h \in \mathcal{H}_{D_s}^*} \mathrm{R}_h^{D_t} [Y \mid Z] \right] \tag{2}$$

where $\mathcal{H}_{D_s}^* := \arg\min_h \mathrm{R}_h^{D_s} [Y \mid Z]$ are the source risk minimizers, and $p_{D_s, D_t}$ is any joint distribution that has full support over $\mathcal{D} \times \mathcal{D}$. We call a representation $Z^*$ (or its encoder) *optimal for IDG* if it minimizes the IDG risk.

### 3.2 CHARACTERIZING OPTIMAL REPRESENTATIONS FOR IDG UNDER COVARIATE SHIFT

The IDG risk is useful to evaluate representations, but gives few insights into IDG and is impractical to optimize due to the supremum in Eq. (2). Under mild assumptions, we provide a simplified, equivalent objective, which is easier to optimize. For convenience, we assume that there is a unique Bayes predictor $f^*$, which minimizes the expected risk over domains, i.e., $f^* = \arg\min_f \mathbb{E}_{p_{D_t}}[\mathrm{R}_f^{D_t} [Y \mid X]]$. This is satisfied by standard ML tasks $p_{Y,X}$ and losses $\ell$. More importantly, we assume the following domain structure, which ensures the existence of optimal encoders and allows our simplification.

**Assumptions.** All domains $d \in \mathcal{D}$ we consider are related by the following assumptions:

1. *Generalized covariate shift.* All domain-specific risk minimizers $f \in \arg\min_f[\mathrm{R}_f^d [Y \mid X]]$ are equal to the Bayes predictor $f^*$ on their support, i.e., $f(x) = f^*(x)$ for all $x \in \mathrm{supp}(p_{X \mid d})$.

2. *Invariance of Bayes predictions.* The set of Bayes predictions is the same for all domains, i.e., $\{f^*(x) \mid x \in \mathrm{supp}(p_{X \mid d})\} = \{f^*(x) \mid x \in \mathcal{X}\}$.

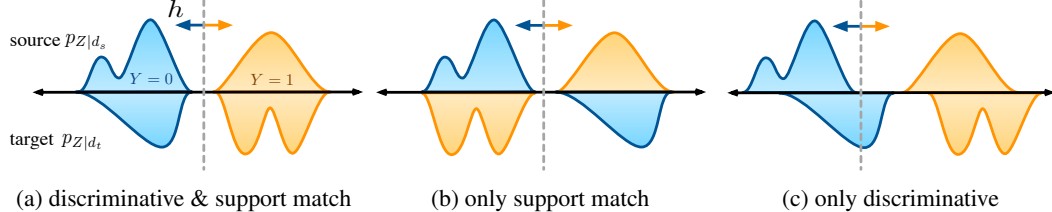

(a) discriminative & support match     (b) only support match     (c) only discriminative

Figure 1: (a) Optimal representations for IDG must have invariant supports while being simultaneously discriminative on all domains: (b) without the discriminative requirement, a source-risk minimizer can mispredict the target, and (c) without support match, some risk minimizer will perform poorly.

Generalized covariate shift (GCS) ensures that $f^*$ is simultaneously optimal on all domains. For log-loss $\ell$ it recovers standard covariate shift, i.e., $p_{Y \mid x,d} = p_{Y \mid x}$. For other losses, GCS is weaker, e.g., it only requires invariance of most likely labels for 0-1 loss, and of conditional expectations for MSE. Invariance of Bayes predictors is necessary to learn useful predictors using a single domain. For example, for 0-1 loss it ensures that each label is seen at least once in each domain.

The intuition behind our objective is that under GCS any source risk minimizer will make optimal predictions on target samples $x$ that are also in the source. Thus, IDG optimal representations are exactly those that (a) have the same support in $\mathcal{Z}$ for all domain, and (b) retain GCS from $Z$ without sacrificing the ability to predict $Y$, which can be ensured by minimizing the risk from $Z$. See Fig. 1.

**Theorem 1.** *Under our assumptions, an encoder $p_{Z^* \mid X}$ is optimal for IDG if and only if it minimizes the risk* $\mathrm{R}\left[Y \mid Z\right] := \inf_h \mathbb{E}_{p_{D_t}}\left[\mathrm{R}_h^{D_t}\left[Y \mid Z\right]\right]$ *while matching the support of $Z$ across domains, i.e.,*

$$p_{Z^* \mid X} \in \underset{p_{Z \mid X}}{\arg\min}\ \mathrm{R}\left[Y \mid Z\right] \quad \text{s.t.} \quad \forall\, d \in \mathcal{D},\ \mathrm{supp}(p_{Z \mid d}) = \mathrm{supp}(p_Z) \tag{3}$$

*Moreover, such encoders exist and their IDG risk is the Bayes risk $\mathrm{R}_{\mathrm{IDG}}\left[Y \mid Z^*\right] = \mathrm{R}\left[Y \mid X\right]$.*

Theorem 1 provides an objective to learn representations on which performing risk minimization using a single domain and $Z^*$ is as good as performing risk minimization on the target domain from inputs $X$. Other sufficient conditions have previously been hinted towards, e.g., matching the marginal $p_{Z \mid d}$ instead of its support (e.g., Ben-David et al., 2010a) which is the focus of most DG methods (e.g., Ganin et al., 2016). Note that previous conditions are nevertheless generally not necessary and could be too stringent to be achievable. To our knowledge, Thm. 1 is the first characterization of necessary and sufficient conditions, which gives better insights into the essential goal for optimal IDG and provides a guide for deriving the *least stringent* objectives in practice.

The risk minimization (Eq. (3)) shows that one must have some knowledge about the target domains to learn optimal representations for IDG. Access to targets might seem unrealistic, but without such knowledge or additional assumptions it is provably impossible to beat even *constant* representations.

**Proposition 1** (No free lunch for IDG). *Let $d_s$ be any source domain, $Z_{d_s}$ be any representation chosen on source $d_s$, and $C \in \mathcal{Z}$ be a constant representation. Under minor assumptions, for every "good" target domain outside the source's support on which $Z_{d_s}$ outperforms $C$ for IDG, there are many "bad" target domains on which $Z_{d_s}$ is strictly worse than $C$. Formal statement in Appx. B.3.*

Proposition 1 shows that target knowledge is necessary for learning useful representations in IDG. This may explain why previous DG methods have been unable to outperform ERM in standard benchmarks (Gulrajani & Lopez-Paz, 2021): the knowledge they have access to is insufficient to generalize. Taken together, Prop. 1 and Thm. 1 say that either you have access to target domains $d_t$, in which case you can achieve an IDG risk that matches supervised learning, or you do not access $d_t$, in which case any representation learning algorithm can achieve worse IDG risk than a constant.

## 4 LEARNING REPRESENTATIONS UNDER COVARIATE SHIFT

### 4.1 SELF-SUPERVISED LEARNING USING DOMAIN-AGNOSTIC AUGMENTATIONS

Our characterization of optimal representations for IDG (Thm. 1) requires labeled data from all domains, which is impractical. We show how this can be overcome with self-supervised learning

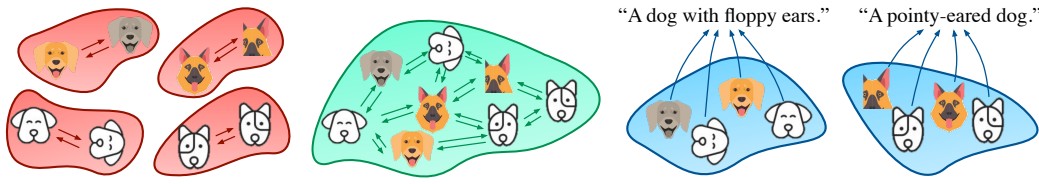

(a) standard augmentations     (b) supervised augmentations     (c) image-text augmentations

Figure 2: Image-text augmentations are practical domain-agnostic augmentations. Arrows denote augmenters. Bubbles denote inputs that have the same representations, as induced by predicting the augmentations. (a) Standard augmentations are not domain-agnostic. (b) Supervised augmentations uniformly augment inputs inside their label class, irrespective of domains. (c) Image-text augmentations are (nearly) domain-agnostic as they map images across domains to similar descriptions.

(SSL), which is a technique for training representations without direct access to labels, and a particular class of data augmentations. E.g, in CLIP, images are augmented with alt-text collected on the internet and invariance is enforced between the representations of the image and its text pair (Radford et al., 2021). Representations learned like this preserve discriminative information about all downstream tasks $Y$ whose label information is preserved by the augmentation (e.g., Dubois et al., 2021).

More precisely, an augmentation $A$ is a random variable sampled conditionally from the input $X$. The key requirement is that augmentations retain task information. Specifically, if any samples $x, x' \in \mathcal{X}$ have the same augmentation conditional $p_{A\,|\,x} = p_{A\,|\,x'}$, then their Bayes predictions must be the same $f^*(x) = f^*(x')$. With such $A$, one can learn an encoder that minimizes the risk $\mathrm{R}\,[Y\,|\,Z]$ by instead maximizing mutual information $\mathrm{I}[A; Z]$. Intuitively, if $Z$ has all augmentation information, then it must have information about the conditional $p_{A\,|\,X}$, and thus the Bayes prediction $f^*(X)$.

This suggests learning optimal representations for IDG by replacing Eq. (3) with a maximization of $\mathrm{I}[A; Z]$. Unfortunately, fully optimizing $\mathrm{I}[A; Z]$ w.r.t. $p_{Z\,|\,X}$ is not generally possible under the support constraint Eq. (3). This can be overcome under a *domain-agnostic* assumption, which requires that the set of possible augmentation distributions is the same across domains, i.e., $\{p_{A\,|\,x}\,|\,x \in \mathrm{supp}(p_{X\,|\,d})\} = \{p_{A\,|\,x}\,|\,x \in \mathcal{X}\}$.

**Proposition 2.** *Let $p_{A\,|\,X}$ be a domain-agnostic augmenter. Then any optimal solution $p_{Z^*\,|\,X}$ of the following objective is optimal for IDG:*

$$p_{Z^*\,|\,X} \in \underset{p_{Z\,|\,X}}{\arg\max}\ \mathrm{I}[A; Z] \quad \text{s.t.} \quad \forall\, d \in \mathcal{D},\ \mathrm{supp}(p_{Z\,|\,d}) = \mathrm{supp}(p_Z) \tag{4}$$

Proposition 2 shows that we can still learn IDG optimal representations without labels if we have access to the right augmentations. How realistic are those augmentations? For 0-1 loss $\ell$, the most likely label should be preserved, which is satisfied by standard image augmentations like rotations and color jittering. Those augmentations are nevertheless not domain-agnostic for typical domains (e.g. sketches and photos), since outputs $A$ are correlated with the input's domain $D$. See Fig. 2a.

A practical choice of augmentation that is nearly domain-agnostic, is a mapping from images to text descriptions, as with CLIP (Radford et al., 2021) which uses text-image pairs. Image-text augmentations have many advantages. First, text augmentations preserve label information for many downstream tasks. Second, they are close to being domain-agnostic, since images from different domains (e.g., sketches and photos) but similar semantics are often mapped to similar descriptions.[2] (Fig. 2c). This gives insights into the open question (Radford et al., 2021) about why CLIP's representations are so robust compared to other SSL methods. Finally, image-text pairs are easy to access in practice given their abundance on the internet. Many other multi-modal augmentations, e.g., audio-video (Wang et al., 2021), are also likely domain-agnostic and can be explored in practice.

In practice, even the domain information $D$ is usually unknown. One can nevertheless still optimize (Eq. (4)) by replace the support constraint with a stronger one that does not rely on $D$ e.g., minimizing $\mathrm{I}[Z; X]$ (see Sec. 4.2.2), . This highlights the potential of Prop. 2: if one can find a large source of inputs $X$ and domain-agnostic augmentations $A$ (e.g., the 400M image-text pairs of CLIP) then one can, in principle, learn optimal representations for IDG on *any* downstream task $Y$ that $A$ preserves.

---

[2]Although text descriptions might contain domain information (e.g., referring to "sketch"), they are still much better than standard augmentations that rarely map together images from different domains.

## 4.2 PRACTICAL OBJECTIVES

We now design practical objectives for learning optimal representations without labels. Proposition 2 does provide an objective but it is impractical as it involves constrained optimization. We can nevertheless convert it to the following unconstrained objective by using a Lagrangian relaxation and introducing a *domain bottleneck* $B[Z, D]$ that enforces support match,

$$\underset{p_{Z \mid X}}{\arg\min} \quad - I[A; Z] + \lambda B[Z, D], \tag{5}$$

Eq. (5) is a valid reformulation of Prop. 2 as long as minimizing $B[Z, D]$ while maximizing $I[A; Z]$ enforces the support constraint in Eq. (4). Below, we provide different choices of such $B[Z, D]$ each of which results in a different SSL objective. In practice, however, terms in Eq. (5) are hard to estimate from finite samples. We now discuss two variational bounds that can be efficiently estimated and optimized with stochastic gradient descent (Bottou, 2010). For simplicity, we use a deterministic encoder $e_\varphi : \mathcal{X} \to \mathcal{Z}$ for the rest of the paper. Detailed derivations are in Appx. C.

For both practical objectives we use a contrastive variational lower bound on $I[A; Z]$ based on InfoNCE (Oord et al., 2018), which is standard in SSL. Specifically, for a sample $X$, we first obtain the augmented 'positive' $A$ by sampling from $p_{A \mid X}$. We then obtain $n$ augmented 'negatives' $\{A_i^-\}_{i=1}^n$ i.i.d. from the marginal $p_A$ by first independently sampling $\mathbf{X} := \{X_i^-\}_{i=1}^n$ from $p_X$ and then sampling $A_i^-$ from $p_{A \mid X_i^-}$. We denote $\mathbf{A} := \{A, A_1^-, \ldots, A_n^-\}$. InfoNCE then uses a critic $s_\psi$ to score how likely each $A' \in \mathbf{A}$ is to be positive, resulting in the following variational bound,

$$I[A; Z] \geq \log(n + 1) + \mathbb{E}_{p_{\mathbf{A}, X, Z}} \left[ \log \frac{\exp s_\psi(A, Z)}{\sum_{A' \in \mathbf{A}} \exp s_\psi(A', Z)} \right]. \tag{6}$$

When $\mathcal{A} = \mathcal{X}$, one can tie the parameters of the critic and the encoder by passing augmentations through the encoder and taking an inner product, i.e., $s_\psi(A, Z) := e_\varphi(A)^T Z$.

Many previous DG regularizers (e.g., Ganin et al., 2016; Li et al., 2018b;a) could be valid domain bottlenecks. In the following, we discuss two possible $B[Z, D]$, the first of which is novel.

### 4.2.1 CONTRASTIVE ADVERSARIAL DOMAIN BOTTLENECK (CAD)

Our first domain bottleneck minimizes $B[Z, D] = I[Z; D]$, which enforces support match using a KL divergence. Dropping constants w.r.t. $Z$ we thus aim to maximize $H[D \mid Z]$. Domain-adversarial neural network (DANN, Ganin et al., 2016) does so by ensuring that a domain classifier $q_\phi$ cannot predict domains from representations, i.e., it maximizes $\mathbb{E}_{p_{D, Z}}[-\log q_\phi(D \mid Z)] \geq H[D \mid Z]$ w.r.t. encoder parameter $\varphi$ but minimizes it w.r.t. $\phi$. However, DANN suffers from two issues: (i) it maximizes an *upper* bound on the desired term; (ii) it requires adversarial training, which is challenging in practice.

---

**Algorithm 1** CAD objective

**Require:** $e_\varphi, s_\psi, D, X, n$
1: $Z \leftarrow e_\varphi(X)$
2: $A \leftarrow \text{sample}(p_{A \mid X})$
3: $\{(D_i^-, X_i^-, A_i^-)\}_{i=1}^n \overset{\text{i.i.d.}}{\longleftarrow} \text{sample}(p_{D, X, A})$
4: $\mathbf{X}, \mathbf{A} \leftarrow \{X_i^-\}_{i=1}^n, \{A\} \cup \{A_i^-\}_{i=1}^n$
5: $\mathbf{X}_{\neg D} \leftarrow \{X_i^- \mid D_i^- \neq D, i \in [n]\}$
6: $\mathcal{L}_{\text{aug}} \leftarrow -\log \frac{\exp s_\psi(A, Z)}{\sum_{A' \in \mathbf{A}} \exp s_\psi(A', Z)}$ $\quad \triangleright -I[A; Z]$
7: $\mathcal{L}_{\text{supp}} \leftarrow -\log \frac{\sum_{X' \in \mathbf{X}_{\neg D}} \exp e_\varphi(X')^T Z}{\sum_{X'' \in \mathbf{X}} \exp e_\varphi(X'')^T Z}$ $\quad \triangleright I[Z; D]$
8: **return** $\mathcal{L}_{\text{CAD}} = \mathcal{L}_{\text{aug}} + \lambda \mathcal{L}_{\text{supp}}$

---

To overcome these issues, we construct $q(D \mid Z)$ without introducing additional parameters and with a bound that is tight with enough samples. In short, using the equality $p_{D \mid Z} = \mathbb{E}_{p_{X \mid Z}}[p_{D \mid X}]$, we set our variational distribution to $q(D \mid Z) = \mathbb{E}_{q_{\varphi, \mathbf{X}}}[\hat{p}(D \mid X)]$, where $q_{\varphi, \mathbf{X}}(X \mid Z)$ is a contrastive variational distribution of $p_{X \mid Z}$ constructed with samples $\mathbf{X}$ and a critic $e_\varphi(X)^T Z$ tied with the encoder, $\hat{p}$ is a count estimate of $p_{D \mid X}$. Detailed derivations and explanations are in Appx. C.3. The resulting contrastive adversarial domain (CAD) objective is in Algorithm 1. First, sample domains $\mathbf{D} := \{D_i^-\}_{i=1}^n$ for each $X' \in \mathbf{X}$. Then collect inputs associated with a different domain from the current domain $D$, i.e., $\mathbf{X}_{\neg D} := \{X_i^- \mid D_i^- \neq D, i \in [n]\}$. Ignoring constants, the final loss is

$$\mathcal{L}_{\text{CAD}}(\varphi, \psi) := \mathbb{E}_{p_{\mathbf{D}, \mathbf{X}, \mathbf{A}, Z}} \left[ -\log \frac{\exp s_\psi(A, Z)}{\sum_{A' \in \mathbf{A}} \exp s_\psi(A', Z)} - \lambda \log \left( \sum_{X' \in \mathbf{X}_{\neg D}} q_{\varphi, \mathbf{X}}(X' \mid Z) \right) \right]. \tag{7}$$

In Appx. C.4, we also derive a conditional variation of CAD that minimizes $I[Z; D \mid Y]$, which can be used when labels are available and supervised augmentations are used.

### 4.2.2 ENTROPY BOTTLENECK (ENT)

Our second domain bottleneck is the entropy bottleneck (Ent) that minimizes $H[Z] = I[Z;X] \geq I[Z;D]$, where the first equality uses the encoder's determinism. Ent enforces support match by removing all information that is not needed to maximize $I[Z;A]$. In particular, minimizing $I[Z;X]$ is more stringent than $I[Z;D]$, as it also matches the representations inside a domain. The advantage of Ent is that it does not require domain samples $\mathbf{D}$, which are rarely accessible in SSL. We consider the standard variational bound used in neural compression (Ballé et al., 2016; Theis et al., 2017), $H[Z] \leq \mathbb{E}_{p_Z}[-\log q_\theta(Z)]$, where an entropy model $q_\theta(Z)$ is used. This leads to

$$\mathcal{L}_{\text{Ent}}(\psi, \varphi, \theta) := \mathbb{E}_{p_{X,\mathbf{A},Z}}\left[-\log\frac{\exp s_\psi(A, Z)}{\sum_{A' \in \mathbf{A}} \exp s_\psi(A', Z)} - \lambda \log q_\theta(Z)\right]. \tag{8}$$

## 5 RELATED WORK

**Provably robust representations under covariate shift.** Previous work mostly focuses on domain generalization bounds for robust representations. Ben-David et al. (2007; 2010a) bound the target risk using the source risk, a divergence between source and target distributions, and the joint optimal risk over source and target domains. Mansour et al. (2009) generalizes these results from 0-1 loss to more general losses. Johansson et al. (2019) takes this further by deriving a support-based bound. In our setting, these bounds only *hint* towards a sufficient condition for optimality, i.e., matching the marginal $p_{Z|d}$ or its support while minimizing $R[Y|Z]$. However, these bounds can often be loose and the implied sufficient conditions are neither necessary nor generally achievable. Ben-David et al. (2010b) suggests that separately minimizing $R[Y|Z]$ or matching the marginal is not sufficient, while Zhao et al. (2019) also proves minimizing only the source risk $R^{d_s}[Y|Z]$ is not sufficient; but none of them proves the desired necessary condition. Our work distinguishes from previous work on three key aspects: (i) we are the first to study and formalize optimally robust representations, and provide the *achievable* sufficient and *necessary* conditions; (ii) we prove that one can practically learn optimal $Z^*$ with SSL using domain-agnostic augmentations; (iii) we consider a more general framework with any standard losses and a less stringent generalized covariate shift assumption, Still, our work is more specific than others, as we consider idealized DG and unrestricted predictors $\mathcal{H}$.

**Practical objectives for DG.** The most popular DG methods aim to learn domain-invariant representation by minimizing various divergernces between the marginal distributions $p_{Z|d}$ and $p_Z$ (Long et al., 2015; Ganin et al., 2016; Sun & Saenko, 2016; Long et al., 2017; Li et al., 2018a; Shen et al., 2018; Nguyen et al., 2021). Others propose matching the conditional $p_{Z|y,d}$ across domains instead (Gong et al., 2016; Li et al., 2018b; Tachet des Combes et al., 2020). These regularizers would all be valid domain bottlenecks $B[Z,D]$. Another line of work aims at learning $Z$ with invariant predictors $p_{Y|z,d}$ across domains (e.g., Arjovsky et al., 2019; Krueger et al., 2021; Li et al., 2021). However, none of these methods outperform ERM with fair model selections (Gulrajani & Lopez-Paz, 2021).

## 6 EXPERIMENTS

In our experiments, we aimed to: (i) verify our theoretical results in practice; (ii) investigate our proposed representation learning objectives in practical DG; (iii) take advantage of pretrained SSL models (in particular, CLIP) to achieve powerful models for DG. Unless stated otherwise, we consider a two-stage training setup. First, the representation learner ("the representor") trains an encoder $p_{Z|X}$ using a specified objective and freezes it. Then, the person performing predictions ("the learner") trains her predictor $h$ from $Z$ by minimizing the risk on source data. Finally, the representation $Z$ and predictor $h$ are evaluated on target data. In all experiments, the learner uses a linear classifier for $h$. For the Ent bottleneck, we used Ballé et al.'s (2018) entropy model. For the CAD bottleneck we used its conditional version whenever labels were available. When a model contains no domain bottleneck, we label it as "Base". For experimental details and additional results see Appxs. E and F.

### 6.1 SCIENTIFIC SETTING: EXPLORING OPTIMAL REPRESENTATIONS FOR WORST-CASE DG

To validate our theory, we studied optimal representations in a scientific setup that is as close to our IDG framework as possible with log-loss $\ell$. In particular, we used the PACS dataset (Li et al., 2017)

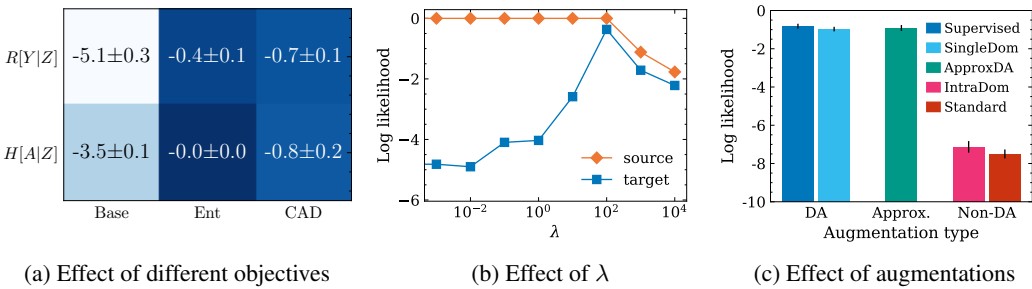

(a) Effect of different objectives    (b) Effect of $\lambda$    (c) Effect of augmentations

Figure 3: (a) Adding bottlenecks significantly improves the worst-case DG performance and using domain-agnostic (DA) augmentations ($\text{H}[A \mid Z]$) performs as well as with labels ($\text{R}[Y \mid Z]$). (b) Increasing the domain bottleneck weight $\lambda$ will improve target performance until it decreases source performance. (c) DA augmentations are crucial but approx. DA aug. might be also be sufficient.

and approximated the *idealized* DG by treating the dataset as the population distribution, i.e., we did not split datasets into train and test sets. To approximate the worst-case source predictor, we followed Dubois et al. (2020) by incorporating the *wrongly labeled target* data to the source domain. The experimental setup goes as follows: (i) the representor trains a ResNet-18 (He et al., 2016) to minimize the objective on labeled data from all domains; (ii) the learner trains a *worst-case* source classifier $h$ on every possible pair of (source, target); (iii) the negative target risk (log likelihood) for each $h$ is evaluated. We reported the log likelihood averaged over 5 seeds. For more realistic scenarios (i.e. non-idealized average-case DG) see Appx. F.2 which replicates the following results.

**Do our domain bottlenecks improve worst-case DG?**    In Fig. 3a, we compare IDG performance of representations trained with (Ent, CAD) and without (Base) domain bottlenecks. We see that both bottlenecks significantly improve the worst-case DG, and nearly achieve the source-domain performance (0 log likelihood). This shows the importance of support match (Thm. 2) and the effectiveness of our bottlenecks to enforce it. In Appx. F.2, we show that bottlenecks also helps in practical scenarios, i.e., non-idealized average-case DG evaluated with accuracy ($95.9\% \to 96.7\%$).

**What is the effect of $\lambda$?**    Fig. 3b shows the effect of the bottleneck weight $\lambda$ on the worst-case target and source performance. We see that increasing $\lambda$ will decrease the DG gap. As a result the target performance improves until $\lambda \approx 10^2$, where source performance starts to decrease.

**What if the representor has access to domain-agnostic augmentations instead of labels?**    In Sec. 4.2, we provide a contrastive objective for using augmentations. To show the effectiveness of the objective, we compared minimizing $\text{H}[A \mid Z]$ using Eq. (6) to standard supervised risk minimization $\text{R}[Y \mid Z]$ and used the domain-agnostic supervised augmentations (Fig. 2b). The 1st and 2nd row of Fig. 3a show that our objective performs similarly to direct label prediction.

**How important is the choice of augmentations?**    Prop. 2 shows that domain-agnostic (DA) augmentations are sufficient for achieving IDG, but it does not give necessary conditions. Here we investigate the effect of using our loss with different choices of augmentations. Specifically, we used $\mathcal{L}_{\text{CAD}}$ with five augmentations. The first two are DA. 'Supervised': augment inputs inside the label class across all domains as in Fig. 2b; 'SingleDom': augment inputs to same label samples from a fixed domain. The second two are not DA. 'Standard': standard SSL augmentations (Chen et al., 2020) as in Fig. 2a; 'IntraDom': augment inputs to same label and same domain samples. Finally, we consider 'ApproxDA', which is approximately DA by augmenting $10\%$ of the time with 'Supervised' and $90\%$ of the time with 'IntraDom'. Fig. 3c shows that the non-DA augmentations give terrible results compared to DA. Interestingly, 'ApproxDA' also performs very well, which suggests that approximately DA augmentations might be sufficient to learn optimal representations in practice.

**What if the representor does not have access to target domains?**    Prop. 1 shows that DG without access to target domains is generally impossible. We empirically verified this by excluding a predefined target $d_t$ domain from the representor's training set, i.e., $\mathcal{L}_{\text{CAD}}$ is optimized on 3 of the 4 domains. The learner then trains a predictor $h$ on each source. We finally evaluate each $h$ on the target domain $d_t$, and average over choices of $d_t$. The resulting worst-case log likelihood was $-4.2 \pm 0.2$, which is significantly worse than when the representor had access to all domains ($-0.8 \pm 0.2$).

## 6.2 APPROXIMATING OPTIMAL REPRESENTATIONS BY EXPLOITING PRETRAINED SSL

As discussed in Sec. 4.1, one can learn optimal representations for IDG by performing SSL with a domain bottleneck on a large sample of inputs $X$ and domain-agnostic augmentations $A$. This is nearly how CLIP was pretrained (SSL with 400M image-text pairs) except it did not include a domain bottleneck. In this section, we investigate how to take advantage of CLIP to approximate optimal representations for IDG. We did so in two simple steps. First, we froze the pretrained CLIP and added a multi-layer perceptron (MLP) that could effectively finetune CLIP's representations. Then, we trained the MLP by minimizing our CAD bottleneck and $\mathrm{R}\left[Y \mid Z\right]$ on the available data.

In all experiments, we used the standard DomainBed benchmark (with non-MNIST datasets) and protocol (Gulrajani & Lopez-Paz, 2021). In particular, we left out a target domain for evaluation and used the union of other domains for training both the encoder and the classifier. Contrary to our scientific setting, the representor does not get access to the target domain. All our representations were evaluated by fitting a linear classifier on source domains with source validation selection. As in DomainBed we selected the encoder based on 'oracle selection' over 10 hyperparameters, and reported the target accuracy averaged over all choices of targets and 5 random seeds with standard errors. Note that using 'oracle selection' is more consistent with our theory since it gets access to the necessary target information (for model selection), as discussed in Appx. F.3. Due to space limit, we only included as baselines 'ERM' and 'DomainBed SOTA' which for *each* dataset is the best result over all baselines. The extended results and baselines are in Table 4. Details in Appx. E.3. We investigated two pretrained CLIP models with different number of parameters. The larger ViT-B/32 denoted 'CLIP L' and the smaller ResNet-50 denoted 'CLIP S'.

Table 1: CLIP significantly outperforms the previous SOTA result on DomainBed, as supported by our theoretical analysis. Finetuning CLIP with our CAD bottleneck consistently improves the robustness of its representations and achieves SOTA performance.

| Algorithm | VLCS | PACS | OfficeHome | TerraIncognita | DomainNet |
|---|---|---|---|---|---|
| ERM | $77.6 \pm 0.3$ | $86.7 \pm 0.3$ | $66.4 \pm 0.5$ | $53.0 \pm 0.3$ | $41.3 \pm 0.1$ |
| DomainBed SOTA | $79.9 \pm 0.2$ | $87.2 \pm 0.1$ | $68.4 \pm 0.2$ | $\mathbf{54.4 \pm 0.3}$ | $41.8 \pm 0.1$ |
| DINO + CAD | $69.6 \pm 0.6$ | $76.1 \pm 0.1$ | $56.9 \pm 0.5$ | $25.9 \pm 1.2$ | $33.6 \pm 0.1$ |
| CLIP S | $81.1 \pm 0.5$ | $90.3 \pm 0.2$ | $70.6 \pm 0.1$ | $29.6 \pm 0.8$ | $47.7 \pm 0.0$ |
| CLIP S + Base | $81.3 \pm 0.5$ | $91.2 \pm 0.3$ | $70.6 \pm 0.1$ | $36.4 \pm 0.7$ | $46.8 \pm 0.2$ |
| CLIP S + CAD | $\mathbf{82.3 \pm 0.3}$ | $92.0 \pm 0.2$ | $71.9 \pm 0.2$ | $36.2 \pm 0.8$ | $48.8 \pm 0.1$ |
| CLIP L | $80.7 \pm 0.4$ | $93.7 \pm 0.8$ | $79.6 \pm 0.1$ | $36.9 \pm 0.6$ | $52.8 \pm 0.1$ |
| CLIP L + CAD | $81.6 \pm 0.1$ | $\mathbf{94.9 \pm 0.3}$ | $\mathbf{80.0 \pm 0.2}$ | $40.6 \pm 1.1$ | $\mathbf{53.7 \pm 0.1}$ |
| Approx. Optimal $Z^*$ | $86.8 \pm 0.6$ | $97.2 \pm 0.6$ | $86.3 \pm 1.6$ | $76.5 \pm 4.1$ | $66.7 \pm 0.2$ |

**Can we approximate optimal representations by exploiting pretrained CLIP?** The row 'CLIP L + CAD' in Table 1 shows that finetuning a large pretrained CLIP model with our CAD achieves SOTA on nearly all DomainBed benchmarks by a very large margin (see 2nd row). Note that the poor performance on TerraIncognita is likely because CLIP's dataset does not cover such images (camera traps monitoring animals). The last row essentially shows an optimal representation, which we approximate by finetuning CLIP L with our CAD on *all* domains including the target. The gap between CLIP L + CAD and the upper-bound suggests that one can still learn better representations. We hypothesize that end-to-end training of our objective would greatly shrink this gap.

**Are gains due to the architectural differences?** DomainBed's baselines finetuned an ImageNet pretrained ResNet-50. In contrast, CLIP L pretrained a larger ViT. To decouple gains due to our objective from architectural gains, we evaluated ResNet-50 pretrained CLIP S. Table 1 shows that CLIP S + CAD still significantly outperforms DomainBed baselines. Note that our theory does not constrain the encoder and so we expect larger encoders to be better as seen in Table 1.

**What is the effect of domain bottlenecks?** In the "CLIP" rows of Table 1, we investigated the effect of finetuning CLIP with our CAD bottleneck. We see that for both CLIP L and CLIP S, it consistently improves results by around $1 \sim 2\%$. These gains are due to the bottleneck, rather than finetuning on source data as seen by 'CLIP S + Base'. We believe the gains could potentially be much larger if CLIP was trained end-to-end with our bottleneck. Note that raw CLIP S already significantly

outperforms baselines. We hypothesize that this is because SGD acts as an information bottleneck that naturally favors support match (Shwartz-Ziv & Tishby, 2017).

**Which pretrained SSL model to use?**    Our theory suggests that we can exploit pretrained SSL models as long as their augmentations are domain-agnostic and their training set covers desired domains. We investigated adaption of SSL models that do not satisfy those properties by finetuning DINO (Caron et al., 2021), the current SOTA on SSL ImageNet. DINO is pretraiend using standard augmentations. As a result, Table 1 shows that the finetuned DINO + CAD significantly underperforms compared to CLIP S and DomainBed baselines. This supports our hypothesis that CLIP is much more robust than other SSL methods due to its domain-agnostic augmentations.

## 6.3    TOWARDS GENERIC ROBUST REPRESENTATIONS WITH SSL

In the previous section, we finetuned CLIP in a task specific fashion by optimizing $R[Y \mid Z]$ and our CAD bottleneck. To get generic (task agnostic) robust representations, one should instead directly use our objectives on a sufficiently large dataset with image-text augmentations. Unfortunately, we cannot fully train CLIP with our bottlenecks as we do not have access to CLIP's original dataset and sufficient compute. In this section, we aim to emulate such training of generic robust representations.

To do so we used LAION-400M (Schuhmann et al., 2021) that is a public dataset that contains 400M web-crawled image-text pairs. Due to our computational budget, we again froze the pretrained CLIP L and only finetuned an additional MLP with our $\mathcal{L}_{\text{Ent}}$. We used $\mathcal{L}_{\text{Ent}}$ as it only requires access to paired image $X$ and text $A$ but no prior information about domain $D$. As in CLIP's paper, we evaluated the learned representation $Z$ in Taori et al.'s (2020) realistic setting, where a linear classifier $h$ from $Z$ is trained on ImageNet and tested on 7 natural distribution shift datasets. Details in Appx. E.4.

**Would training CLIP with a bottleneck have improved its robustness?**    As shown in the last 2 rows of Table 2, finetuning CLIP L on LAION with $\mathcal{L}_{\text{Ent}}$ (Tuned w/ Ent) outperforms finetuning without bottleneck (Tuned w/o Ent) on all 7 distribution shift datasets. This suggests that directly training CLIP with our Ent bottleneck would improve the robustness of learned representations. We hypothesize that the gains could be larger if SSL models trained $\mathcal{L}_{\text{Ent}}$ end-to-end. In Appx. F.4, we show similar results on DomainBed. Note that both models underperform the original CLIP L, likely due to non-end-to-end training and LAION data with (possibly) lower quality than CLIP's data.

Table 2: Finetuning CLIP L on LAION with an entropy bottleneck improves its robustness compared to finetuning without on 7 distribution shift datasets. The pretrained CLIP L is still better likely due to end-to-end training with higher quality data. IN denotes ImageNet.

|  | IN | IN-V2 | IN-S | YT-BB | IN-Vid | ObjectNet | IN-A | IN-R | Avg. |
|---|---|---|---|---|---|---|---|---|---|
| CLIP L | 75.2 | 64.2 | 41.0 | 58.4 | 71.6 | 42.8 | 27.5 | 62.9 | 52.6 |
| Tuned w/o Ent | 73.8 | 62.1 | 37.0 | 56.9 | 68.8 | 41.3 | 26.0 | 58.1 | 50.0 |
| Tuned w/ Ent | 74.2 | 62.7 | 38.9 | 58.1 | 70.1 | 42.1 | 26.2 | 60.8 | 51.3 |

## 7    CONCLUSION

We gave a simple variational characterization of all representations on which source-risk minimizers are guaranteed to generalize to target domains that preserve the Bayes predictor. Similar to previous work, our theory strongly implies the need for target information when learning representations for domain generalization. Nevertheless, we identified a domain-agnostic property of data augmentations that make it possible to learn optimal representations from unlabelled data. Thus, we showed that it is possible to learn robust representations using only large sources of inputs $X$ and augmentations $A$.

There are caveats that need to be addressed in future work. First, we studied an idealized DG, which assumes access to the population distributions. This gives insights into the challenges that are specific to DG, rather than finite sample challenges faced throughout ML. Second, we considered risk minimizers from an unconstrained hypothesis class. The support constraint can likely be weakened, if the hypothesis class is constrained. Finally, we focus only on optimal representations, but it would be interesting to characterize approximately optimal representations. Nevertheless, in this idealized setting, our characterization is a springboard from which all future objectives can be derived, and, in general, it brings us closer to the goal of robust machine learning systems.

**Acknowledgement**   We would like to thank Elliot Creager, Roger Grosse, Elan Rosenfeld, Guodong Zhang, Han Zhao, and anonymous reviewers for their helpful feedbacks and encouragements. Resources used in preparing this research were provided, in part, by the Province of Ontario, the Government of Canada through CIFAR, and companies sponsoring the Vector Institute. We acknowledge the support of the Natural Sciences and Engineering Research Council of Canada (NSERC), RGPIN-2021-03445.

**Reproducibility**   For our theoretical results, we include formal assumptions, statements, and proofs in Appxs. A and B. We include the detailed derivations of our algorithms in Appx. C. For our experiments, we include experimental details for reproducing our results in Appx. E and have released our code at `https://github.com/ryoungj/optdom`.

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

# Appendix

## Table of Contents

## A  PRELIMINARIES

### A.1  NOTATION

For the most part, we will assume that all spaces are discrete probability spaces. A full list of assumptions is found at Appx. A.3.

**General**  The image of a set $A \subseteq \mathcal{X}$ under a function $f : \mathcal{X} \to \mathcal{Y}$ is denoted $f^{\to}(A) = \{f(x) \mid x \in A\}$. The pre-image is denoted $f^{\leftarrow}(B) = \{x \in \mathcal{X} \mid f(x) \in B\}$ for $B \subseteq \mathcal{Y}$.

**Probability**  Random variables (r.v.) are denoted by uppercase letters (e.g., $X$), and their sample space and realizations are denoted by the corresponding calligraphic (e.g., $\mathcal{X}$) and lowercase letters (e.g., $x$) respectively. The probability mass function (pmf) of a random variable $X$ is denoted as $p_X$. We use capital $P$ instead of $p$ to denote the measure under $p$. The support $\mathrm{supp}(p_X)$ of a discrete distribution is the set of all points $x \in \mathcal{X}$ with positive probability, i.e., $\mathrm{supp}(p_X) = \{x \in \mathcal{X} \mid p_X(x) > 0\}$. The space of all probability distributions on $\mathcal{X}$ is denoted $\mathcal{P}(\mathcal{X}) = \{p_X \mid p_X(x) \geq 0 \text{ and } \sum_{x \in \mathcal{X}} p_X(x) = 1\}$.

When it is necessary to be explicit, we will denote '$X$ is distributed as $p_X$' using the notation $X \overset{\mathrm{d}}{\sim} p_X$. Expectations are written as: $\mathbb{E}_{p_X}[f(X)]$, independence of two r.v. as $\cdot \perp\!\!\!\perp \cdot$, conditional independence as $\cdot \perp\!\!\!\perp \cdot \mid \cdot$.

For jointly distributed random variables $(X, Y)$ taking value in (t.v.i.) $\mathcal{X} \times \mathcal{Y}$, the conditional distribution is denoted as $p_{Y \mid X} : \mathcal{Y} \times \mathcal{X} \to [0, 1]$. For convenience, let $p_{Y \mid x} = p_{Y \mid X}(\cdot \mid x)$ be the conditional distribution of $Y$ given $x$. All random variables are independently distributed, unless an explicit joint distribution or coupling is given.

### A.2  DEFINITIONS

We are interested in prediction problems with domain shift. There are three random variables: the target domain $D_t$, the input $X$, the label $Y$. They have the following joint distribution:
$$(D_t, X, Y) \overset{\mathrm{d}}{\sim} p_{D_t} \cdot p_{X, Y \mid D_t} \tag{9}$$
where we drop the arguments of the probability densities for clarity. We make a variety of convenience assumptions on these random variables (Assumption 6). Crucially, we will be making the Bayes invariance assumption on $p_{D_t, X, Y}$ that can be thought of as a generalized covariate shift assumption (Assumption 4).

We will be studying the effect of changing the representation of the data. This is done by "encoding" $X$ into a representation $Z$ using a conditional distribution $p_{Z \mid X}$.

**Definition 1** (Encoder). An *encoder* is a conditional distribution $p_{Z \mid X} : \mathcal{Z} \times \mathcal{X} \to [0, 1]$ from the input space $\mathcal{X}$ to the representation space $\mathcal{Z}$.

The data together with the representation has the following joint:
$$(D_t, X, Y, Z) \overset{\mathrm{d}}{\sim} p_{D_t} \cdot p_{X, Y \mid D_t} \cdot p_{Z \mid X} \tag{10}$$
The key thing to notice here is that $Z$ is conditionally independent of $Y, D_t$ given $X$. In particular, the same encoder is used across all domains.

#### A.2.1  RISK MINIMIZATION

Our ultimate goal is to predict $Y$ from the representation $Z$ of $X$ in a manner that is robust to changes in the domain.

We formalize this in the standard way by making predictions $\gamma \in \Gamma$ in a space of predictions or actions. For example the prediction space may be the set of all possible labels $\Gamma = \mathcal{Y}$, in which case we would be predicting deterministic labels. Or we may predict a distribution over labels, in which case the prediction space would be the set of all probability distributions on $\mathcal{Y}$, i.e. $\Gamma = \mathcal{P}(\mathcal{Y})$.

A *predictor* is a function mapping inputs to predictions, i.e., $f : \mathcal{X} \to \Gamma$, or representations to predictions, i.e., $h : \mathcal{Z} \to \Gamma$. For example, $f$ may be a neural network that takes as input a sample $x$ and outputs a vector of logits that parameterize a softmax distribution over finitely many labels.

We select predictors according to the *risk* defined via a loss function $\ell : \mathcal{Y} \times \Gamma \to \mathbb{R}_{\geq 0} \cup \{\infty\}$:

$$\mathrm{R}_f\left[Y \mid X\right] := \mathbb{E}_{p_{X,Y}}\left[\ell(Y, f(X))\right]. \tag{11}$$

In particular, we are interested in the Bayes (minimum) risk over all predictors:

$$\mathrm{R}\left[Y \mid X\right] := \inf_f \mathrm{R}_f\left[Y \mid X\right], \tag{12}$$

We denote the set of all optimal predictors from $X$ as

$$\mathcal{F}^* := \{f \mid \mathrm{R}_f\left[Y \mid X\right] = \mathrm{R}\left[Y \mid X\right]\} \tag{13}$$

Similarly, we define the risk $\mathrm{R}_h\left[Y \mid Z\right]$, the Bayes risk $\mathrm{R}\left[Y \mid Z\right]$, and the set of optimal predictors

$$\mathcal{H}_z^* := \{h \mid \mathrm{R}_h\left[Y \mid Z\right] = \mathrm{R}\left[Y \mid Z\right]\} \tag{14}$$

from $Z$, all of which vary as a function of the encoder $p_{Z \mid X}$. Note, in the main body of the paper, we omitted the subscript $Z$ from $\mathcal{H}_z^*$ for clarity, but we will keep it in the Appendices. We assume that together our loss and prediction space always admit optima (Item 2 of Assumption 2), and thus $\mathcal{F}^*, \mathcal{H}_z^*$ are always non-empty.

We will be assuming that the risk admits unique optimal prediction when predicting from $X$ (Item 3 of Assumption 2). Thus it makes sense to define the following:

**Definition 2** (The Bayes predictor). *The Bayes predictor $f^* : \mathcal{X} \to \Gamma$ is the unique predictor that is optimal for all $x \in \mathcal{X}$:*

$$f^*(x) = \arg\min_{\gamma \in \Gamma} \mathbb{E}_{p_{Y \mid x}}\left[\ell(Y, \gamma)\right] \tag{15}$$

**Definition 3** (The Bayes image). *The image of all the inputs under the Bayes predictor will be denoted as $\Gamma^* = f^{*\to}(\mathcal{X})$ and called* the Bayes image.

Note that $\mathcal{F}^*$ becomes a singleton $\{f^*\}$, but it is not necessarily the case for $\mathcal{H}_z^*$ since we will not be making any uniqueness assumption on optimal prediction from $Z$.

### A.2.2 DOMAIN GENERALIZATION

We are interested in controlling the risk in a domain generalization setting, and so we define the *domain-conditional risk*,

$$\mathrm{R}_f^d\left[Y \mid X\right] := \mathbb{E}_{p_{X,Y \mid d}}\left[\ell(Y, f(X))\right]. \tag{16}$$

$\mathrm{R}^d\left[Y \mid X\right], \mathcal{F}_d^*$ are defined as Eqs. (12) and (13), respectively, but with respect to $\mathrm{R}_f^d$. Similarly, define the Bayes image for domain $d$ as

$$\Gamma_d^* := f^{*\to}\left(\mathrm{supp}(p_{X \mid d})\right). \tag{17}$$

We also define domain-conditional quantities for prediction from a representation $Z$. The most important term which we will be investigating is an idealization of the domain generalization worst-case risk.

**Definition 4** (IDG risk). *Given an encoder $p_{Z \mid X}$ and a distribution $p_{D_t, D_s}$ over a target domain $D_t$ and source domain $D_s$, the idealized domain generalization worst-case risk, IDG risk for short, is the expected worst-case target risk taken over source minimizers, i.e.,*

$$\mathrm{R}_{\mathrm{IDG}}\left[Y \mid Z\right] := \mathbb{E}_{p_{D_t, D_s}}\left[\sup_{h \in \mathcal{H}_{Z, D_s}^*} \mathrm{R}_h^{D_t}\left[Y \mid Z\right]\right] \tag{18}$$

Note that the IDG risk is well-defined because $\mathcal{H}_{Z, D_s}^*$ is non-empty by Assumption 2. The desired optimal representations, are then those that minimize the IDG risk.

**Definition 5** (Optimal representations for IDG). *An encoder $p_{Z^* \mid X}$ is* optimal *for idealized domain generalization if and only if it minimizes the IDG risk, i.e.,*

$$\mathrm{R}_{\mathrm{IDG}}\left[Y \mid Z^*\right] = \inf_{p_{Z \mid X}} \mathrm{R}_{\mathrm{IDG}}\left[Y \mid Z\right] \tag{19}$$

## A.3 ASSUMPTIONS

We make a the following assumptions throughout the paper. **All these assumptions should hold for practical settings.**

**Assumption 1** (Convenience: discrete probability spaces). All data spaces $(\mathcal{D}, \mathcal{X}, \mathcal{Y}, \mathcal{Z}, \mathcal{A})$ are discrete spaces. Because the distributions of $X, Y, D$ are fixed, we assume for convenience that $\operatorname{supp}(p_X) = \mathcal{X}$, $\operatorname{supp}(p_Y) = \mathcal{Y}$, and $\operatorname{supp}(p_{D_t}) = \mathcal{D}$.

Assumption 1 is a convenience assumption to avoid measure theory for the sake of clarity. It always holds in practice due to finiteness of computers, i.e., all spaces will be finite but arbitrarily large. We believe that our claims can nevertheless be generalized to typical continuous spaces with some minor technical assumptions.

**Assumption 2** (Losses admit optima). We assume that our risk always admits optimal predictions:

1. $|\Gamma| > 1$.

2. For all $p_\Upsilon \in \mathcal{P}(\mathcal{Y})$, there exists $\gamma^* \in \Gamma$, such that
$$\mathbb{E}_{p_\Upsilon}[\ell(\Upsilon, \gamma^*)] \leq \mathbb{E}_{p_\Upsilon}[\ell(\Upsilon, \gamma)] \qquad \forall\, \gamma \in \Gamma. \tag{20}$$

3. For all $x \in \mathcal{X}$, there exist $\gamma^* \in \Gamma$, such that
$$\mathbb{E}_{p_{Y \mid x}}[\ell(Y, \gamma^*)] < \mathbb{E}_{p_{Y \mid x}}[\ell(Y, \gamma)] \qquad \forall\, \gamma \neq \gamma^*. \tag{21}$$

Note that for log-loss $\ell(y, \gamma) = -\log \gamma(y)$ and finite $\mathcal{Y}$, these assumptions are satisfied if $\Gamma = \mathcal{P}(\mathcal{Y})$ where the optimal prediction for Item 3 is $\gamma^* = p_{Y \mid x}$ by strict properness (Gneiting & Raftery, 2007). If we consider the 0-1 loss (reverse accuracy) $\ell(y, \gamma) = 1 - \mathbb{1}[y = \gamma]$ with $\Gamma = \mathcal{Y}$ and a finite label space where the optimal prediction for Item 3 is $\gamma^* = \arg\max_{y \in \mathcal{Y}} p_{Y \mid x}(y)$, this assumption is mostly satisfied, except we assume that $p_{Y \mid x}$ has a unique mode.

Assumption 2 serves two purposes: Item 2 ensures that for any representation the optimal predictors from $Z$ exists such that the IDG risk is well-defined as in Def. 5; Item 3 ensures a unique Bayes predictor from $X$, which simplifies the analysis and is satisfied by common losses as described above.

**Assumption 3** (Cardinalities). We assume that
$$|\mathcal{Z}| \geq |\Gamma^*| \geq 2 \tag{22}$$

Assumption 3 is very weak and ensures that optimal representations always exists (Prop. 3).

**Assumption 4** (Generalized covariate shift). The Bayes predictor is optimal for all domains. I.e., for all $(x, d) \in \operatorname{supp}(p_{X, D_t}), \gamma \in \Gamma$ such that $\gamma \neq f^*(x)$, we have
$$\mathbb{E}_{p_{Y \mid x, d}}[\ell(Y, f^*(x))] < \mathbb{E}_{p_{Y \mid x, d}}[\ell(Y, \gamma)]. \tag{23}$$

For example, in the case of strictly proper scoring rules, e.g. log loss, covariate shift $p_{Y \mid X, D} = p_{Y \mid X}$ is equivalent to the invariance of the Bayes predictor. For the 0-1 loss, this is guaranteed by invariance of the most likely label. For MSE it is guaranteed by the invariance of the expected label. In the latter two cases, Assumption 4 is less stringent than the typical covariate shift assumption.

Assumption 4 is the core assumption for our theoretical results. It ensures that source and target domains are related in a useful way that can be utilized by the representation.

**Assumption 5** (Constant Bayes image). The Bayes image is invariant across domains, i.e., for all $d \in \mathcal{D}$,
$$\Gamma_d^* = \Gamma^*. \tag{24}$$

For the case of 0-1 loss, this simply means that the label set for all domains is the same, which is trivial. For log-loss, this means that the set of possible conditional distributions $\Gamma_d^* = \{p_{Y \mid x} \mid x \in \operatorname{supp}(p_{X \mid d})\}$ is the same across domains.

Assumption 5 is crucial to be able to learn. Without it, in the extreme case, one could set each domain to be all examples associated with a single element from the label set (or the Bayes image set) in which case it is impossible to generalize across different domains. Assumption 5 is also necessary to guarantee the existence of optimal representations as in Prop. 3.

**Assumption 6** (Domain joint). $p_{D_t, D_s}$ is any distribution such that $\mathrm{supp}(p_{D_t, D_s}) = \mathcal{D} \times \mathcal{D}$.

In a simplified scenario, one could define the source $D_s$ and target $D_t$ as i.i.d. r.v. from $p_{D_t}$, where $p_{D_t, D_s} = p_{D_t} \cdot p_{D_s} = p_{D_t} \cdot p_{D_t}$ and Assumption 6 is trivially satisfied.

## B  PROOFS

### B.1  LEMMAS FOR GENERAL LOSSES

An important result that we will be using is the generalized data processing inequality of Bayes risk (Xu & Raginsky, 2020; Dubois et al., 2021). We include it here for completeness.

**Lemma 1** (Generalized DPI (Xu & Raginsky, 2020; Dubois et al., 2021)). *Let $Z - X - Y$ be a Markov chain of random variables. For any loss function $\ell$,*

$$\mathrm{R}\left[Y \mid X\right] \le \mathrm{R}\left[Y \mid Z\right]. \tag{25}$$

For the case of strictly proper losses (Assumption 2) we can go one step further.

**Lemma 2.** *Let $Z - X - Y$ be a Markov chain of random variables. Then, under Assumptions 1 and 2 we have that*

$$\mathrm{R}\left[Y \mid Z\right] = \mathrm{R}\left[Y \mid X\right] \iff \forall h^* \in \mathcal{H}_z^*, \forall (x, z) \in \mathrm{supp}(p_{X,Z}),\ h^*(z) = f^*(x). \tag{26}$$

*Proof.* Suppose that for all $h^* \in \mathcal{H}_z^*$ we have $h^*(z) = f^*(x)$ on the support of $p_{X,Z}$. Then,

$$\mathrm{R}\left[Y \mid X\right] = \mathbb{E}_{p_{X,Y}}[\ell(Y, f^*(X))] \tag{27}$$
$$= \mathbb{E}_{p_{X,Y}p_{Z \mid X}}[\ell(Y, f^*(X))] \tag{28}$$
$$= \mathbb{E}_{p_{X,Y}p_{Z \mid X}}[\ell(Y, h^*(Z))] \tag{29}$$
$$= \mathbb{E}_{p_{Z,Y}}[\ell(Y, h^*(Z))] \tag{30}$$
$$= \mathrm{R}\left[Y \mid Z\right]. \tag{31}$$

Now suppose there exists a $h^* \in \mathcal{H}_z^*$ and a pair $(x', z') \in \mathrm{supp}(p_{X,Z})$ such that $h^*(z') \neq f^*(x')$. Then

$$\mathrm{R}\left[Y \mid Z\right] \tag{32}$$
$$= \mathbb{E}_{p_{X,Z}p_{Y \mid X}}[\ell(Y, h^*(Z))] \tag{33}$$
$$= p_{X,Z}(x', z')\, \mathbb{E}_{p_{Y \mid x'}}[\ell(Y, h^*(z'))] + \sum_{(x,z) \neq (x', z')} p_{X,Z}(x, z)\, \mathbb{E}_{p_{Y \mid x}}[\ell(Y, h^*(z))] \tag{34}$$
$$\ge p_{X,Z}(x', z')\, \mathbb{E}_{p_{Y \mid x'}}[\ell(Y, h^*(z'))] + \sum_{(x,z) \neq (x', z')} p_{X,Z}(x, z)\, \mathbb{E}_{p_{Y \mid x}}[\ell(Y, f^*(x))] \tag{35}$$
$$> p_{X,Z}(x', z')\, \mathbb{E}_{p_{Y \mid x'}}[\ell(Y, f^*(x'))] + \sum_{(x,z) \neq (x', z')} p_{X,Z}(x, z)\, \mathbb{E}_{p_{Y \mid x}}[\ell(Y, f^*(x))] \tag{36}$$
$$= \mathrm{R}\left[Y \mid X\right] \tag{37}$$

Eq. (35) follows by Item 3 of Assumption 2 along with the definition of $f^*$. Eq. (36) follows by Item 3 of Assumption 2 and the fact that $h^*(z') \neq f^*(x')$. This completes the proof, because Lemma 1 prevents $\mathrm{R}\left[Y \mid Z\right] < \mathrm{R}\left[Y \mid X\right]$. $\square$

### B.2  PROOF OF THEOREM 1

First we will show that the desired representation exists by taking all inputs for which the Bayes predictor predicts similarly and "bucketing" them to the same representation. This is a direct extension of the example from Dubois et al.'s (2020) Proposition 6, to the case of proper losses.

**Proposition 3** (Existence of optimal representations). *Under Assumptions 1 to 5, there exists an encoder $p_{Z^* \mid X}$ that is optimal for Eq. (3), i.e.,*

$$p_{Z^* \mid X} \in \arg\min_{p_{Z \mid X}} \mathrm{R}\left[Y \mid Z\right] \quad \text{s.t.} \quad \forall d \in \mathcal{D},\ \mathrm{supp}(p_{Z \mid d}) = \mathrm{supp}(p_Z). \tag{38}$$

*Moreover, we have that*

$$\mathrm{R}\left[Y \mid X\right] = \mathrm{R}\left[Y \mid Z^*\right]. \tag{39}$$

*Proof.* Because we assume arbitrary encoders $p_{Z\,|\,X}$, the essence of this construction is simple: we embed the Bayes image into $\mathcal{Z}$. Indeed, let $\phi : \Gamma^* \to \mathcal{Z}$ be any one-to-one function, which exists due to Assumption 3 (here we use deterministic one-to-one function for simplicity, the construction can be easily extended to stochastic case). Then let $Z^* = \phi(f^*(X))$. We now verify the properties of $p_{Z^*\,|\,X}$.

1. $Z^*$ satisfies $\mathrm{R}\,[Y\,|\,X] = \mathrm{R}\,[Y\,|\,Z^*]$. Indeed,

$$\mathrm{R}\,[Y\,|\,X] = \mathbb{E}_{p_{X,Y}}[\ell(Y, f^*(X))] \tag{40}$$

$$= \mathbb{E}_{p_{X,Y}\,p_{Z^*\,|\,X}}[\ell(Y, f^*(X))] \tag{41}$$

$$= \mathbb{E}_{p_{Z^*,Y}}\left[\ell(Y, \phi^{-1}(Z^*))\right] \tag{42}$$

$$\geq \mathrm{R}\,[Y\,|\,Z^*]. \tag{43}$$

Eq. (42) is by our construction of $Z^*$ and Eq. (43) is by the definition of the Bayes risk. Due to the data processing inequality of Bayes risk (Lemma 1) we also have $\mathrm{R}\,[Y\,|\,X] \leq \mathrm{R}\,[Y\,|\,Z^*]$, from which we conclude that $\mathrm{R}\,[Y\,|\,X] = \mathrm{R}\,[Y\,|\,Z^*]$ and that Eq. (39) holds.

2. Recall that $\Gamma^* = f^{*\to}(\mathcal{X})$ and $\Gamma_d^* = f^{*\to}\big(\mathrm{supp}(p_{X\,|\,d})\big)$. Now let us compute the desired support for all $d \in \mathcal{D}$:

$$\mathrm{supp}(p_{Z^*\,|\,d}) = \phi^{\to}(\Gamma_d^*) \tag{44}$$

$$= \phi^{\to}(\Gamma^*) \tag{45}$$

$$= \mathrm{supp}(p_{Z^*}). \tag{46}$$

Eq. (45) is by Assumption 5.

Because $\mathrm{R}\,[Y\,|\,X]$ is the minimum achievable risk by any encoder regardless of constraint (this is by Lemma 1), this implies that $p_{Z^*\,|\,X}$ is an optimal encoder for Eq. (3). $\qquad\square$

The following lemma essentially says that when $\mathrm{R}\,[Y\,|\,Z]$ is minimized, then the optimal predictors for each domain all agree on the intersection of their support.

**Lemma 3.** *Let $p_{Z\,|\,X}$ be an encoder such that $\mathrm{R}\,[Y\,|\,Z] = \mathrm{R}\,[Y\,|\,X]$. Under Assumptions 1 and 2, we have that for all $z \in \mathrm{supp}(p_Z)$, there exists $\gamma^* \in \Gamma$ such that*

$$\mathbb{E}_{p_{Y\,|\,z}}[\ell(Y, \gamma^*)] < \mathbb{E}_{p_{Y\,|\,z}}[\ell(Y, \gamma)] \qquad \forall\, \gamma \neq \gamma^*. \tag{47}$$

*In other words, the restriction of any $h^* \in \mathcal{H}_z^*$ to $\mathrm{supp}(p_Z)$ is unique. If, in addition, Assumption 4 holds, then for all $(z, d) \in \mathrm{supp}(p_{Z,D_t}), \gamma \in \Gamma$ such that $\gamma \neq h^*(z)$,*

$$\mathbb{E}_{p_{Y\,|\,z,d}}[\ell(Y, h^*(z)] < \mathbb{E}_{p_{Y\,|\,z,d}}[\ell(Y, \gamma)]. \tag{48}$$

*In other words, the restriction of any $h \in \mathcal{H}_{Z,d}^*$ to $\mathrm{supp}(p_{Z\,|\,d})$ is unique and equal to $h^*$.*

*Proof.* For the first result, let $z \in \mathrm{supp}(p_Z)$ and consider $x \in \mathrm{supp}(p_{X\,|\,z})$. By Lemma 2, it must be the case that $f^*$ is constant on $\mathrm{supp}(p_{X\,|\,z})$. Thus, we can pick $\gamma^* = f^*(x)$. Now, let $\gamma \neq \gamma^*$. We have that,

$$\mathbb{E}_{p_{Y\,|\,z}}[\ell(Y, \gamma^*)] = \mathbb{E}_{p_{X\,|\,z}\,p_{Y\,|\,X}}[\ell(Y, \gamma^*)] \tag{49}$$

$$= \mathbb{E}_{p_{X\,|\,z}\,p_{Y\,|\,X}}[\ell(Y, f^*(X))] \tag{50}$$

$$< \mathbb{E}_{p_{X\,|\,z}\,p_{Y\,|\,X}}[\ell(Y, \gamma)] \tag{51}$$

$$= \mathbb{E}_{p_{Y\,|\,z}}[\ell(Y, \gamma)]. \tag{52}$$

Eq. (49) is due to the conditional independence of $Y$ and $Z$ given $X$. Eq. (51) is due to Assumption 2 and the definition of the Bayes predictor. Let $h^* : \mathrm{supp}(p_Z) \to \Gamma$ be the unique Bayes predictor from $Z$.

Now, for the second result, note that

$$\mathrm{R}\,[Y\,|\,X] = \mathrm{R}_{f^*}\,[Y\,|\,X] \tag{53}$$

$$= \sum_{d \in \mathcal{D}} p_{D_t}(d) \, \mathrm{R}_{f^*}^d \left[ Y \mid X \right] \tag{54}$$

$$= \sum_{d \in \mathcal{D}} p_{D_t}(d) \, \mathrm{R}^d \left[ Y \mid X \right], \qquad \text{Assumption 4} \tag{55}$$

and

$$\mathrm{R} \left[ Y \mid Z \right] = \mathrm{R}_{h^*} \left[ Y \mid Z \right] \tag{56}$$

$$= \sum_{d \in \mathcal{D}} p_{D_t}(d) \, \mathrm{R}_{h^*}^d \left[ Y \mid Z \right] \tag{57}$$

$$\geq \sum_{d \in \mathcal{D}} p_{D_t}(d) \, \mathrm{R}^d \left[ Y \mid Z \right], \tag{58}$$

where Eq. (58) is due to the definition of (domain-conditional) Bayes risk. Then

$$\mathrm{R} \left[ Y \mid Z \right] - \mathrm{R} \left[ Y \mid X \right] \geq \sum_{d \in \mathcal{D}} p_{D_t}(d) \left( \mathrm{R}^d \left[ Y \mid Z \right] - \mathrm{R}^d \left[ Y \mid X \right] \right) \tag{59}$$

$$\geq 0. \qquad \text{Lemma 1 conditioned on } d \tag{60}$$

Thus, any encoder that achieves $\mathrm{R} \left[ Y \mid Z \right] = \mathrm{R} \left[ Y \mid X \right]$ also satisfies $\mathrm{R}^d \left[ Y \mid Z \right] = \mathrm{R}^d \left[ Y \mid X \right]$ for all $d \in \mathcal{D}$ since we assume that $\mathrm{supp}(p_{D_t}) = \mathcal{D}$ in Assumption 1. Now, let $d \in \mathcal{D}$. An argument analogous to Lemma 2 gives us,

$$\forall h \in \mathcal{H}_{Z,d}^*, \forall (x, z) \in \mathrm{supp}(p_{X,Z \mid d}), \; h(z) = f^*(x) = h^*(z). \tag{61}$$

Eq. (61) is derived from $\mathrm{R}^d \left[ Y \mid Z \right] = \mathrm{R}^d \left[ Y \mid X \right]$ using Assumption 4 in place of Item 3 of Assumption 2 for a specific domain $d$. Let $z \in \mathrm{supp}(p_{Z \mid d})$ and $\gamma \in \Gamma$ such that $\gamma \neq h^*(z)$. Since $\mathrm{supp}(p_{X \mid z,d}) \subseteq \mathrm{supp}(p_{X \mid z})$, $f^*$ is a constant on $\mathrm{supp}(p_{X \mid z,d})$ and equal to $h^*$. Now, as above, we have that

$$\mathbb{E}_{p_{Y \mid z,d}}[\ell(Y, h^*(z))] = \mathbb{E}_{p_{X \mid z,d} p_{Y \mid X,d}}[\ell(Y, h^*(z))] \tag{62}$$

$$= \mathbb{E}_{p_{X \mid z,d} p_{Y \mid X,d}}[\ell(Y, f^*(X))] \tag{63}$$

$$< \mathbb{E}_{p_{X \mid z,d} p_{Y \mid X,d}}[\ell(Y, \gamma)] \tag{64}$$

$$= \mathbb{E}_{p_{Y \mid z,d}}[\ell(Y, \gamma)]. \tag{65}$$

Eq. (64) is due to Assumption 4. $\qquad \square$

**Corollary 1.** *Let $p_{Z \mid X}$ be an encoder such that $\mathrm{R} \left[ Y \mid Z \right] = \mathrm{R} \left[ Y \mid X \right]$. Under Assumptions 1, 2 and 4 we have that $\mathcal{H}_Z^* \subseteq \mathcal{H}_{Z,d}^*$ for all $d \in \mathcal{D}$ and that for all $d_s, d_t \in \mathcal{D}$*

$$\inf_{h \in \mathcal{H}_{Z,d_s}^*} \mathrm{R}_h^{d_t} \left[ Y \mid Z \right] = \mathrm{R}^{d_t} \left[ Y \mid Z \right] \tag{66}$$

*Proof.* $\mathcal{H}_Z^* \subseteq \mathcal{H}_{Z,d}^*$ is immediate from Lemma 3. Now, we have that $\mathrm{R}_h^{d_t} \left[ Y \mid Z \right] \geq \mathrm{R}^{d_t} \left[ Y \mid Z \right]$. So, the result follows by taking any $h \in \mathcal{H}_Z^* \subseteq \mathcal{H}_{Z,d_s}^*$ in the inf of Eq. (66). $\qquad \square$

**Theorem 2** (Characterizing optimal representations for IDG, equiv. Theorem 1). *Under Assumptions 1 to 6, an encoder $p_{Z \mid X}$ is optimal for idealized domain generalization if and only if it minimizes the Bayes risk while matching the support of $p_{Z \mid d}$ and $p_Z$ for all $d \in \mathcal{D}$, i.e.,*

$$p_{Z \mid X} \in \arg \min_{p_{Z \mid X}} \mathrm{R} \left[ Y \mid Z \right] \tag{67}$$

$$\text{s.t.} \quad \forall \, d \in \mathcal{D}, \; \mathrm{supp}(p_{Z \mid d}) = \mathrm{supp}(p_Z) \tag{68}$$

*Proof.* The IDG risk is lower bounded by $\mathrm{R} \left[ Y \mid X \right]$:

$$\mathrm{R}_{\mathrm{IDG}} \left[ Y \mid Z \right] \geq \mathbb{E}_{p_{D_s, D_t}} \left[ \inf_{h \in \mathcal{H}_{Z,D_s}^*} \mathrm{R}_h^{D_t} \left[ Y \mid Z \right] \right] \tag{69}$$

$$\geq \mathbb{E}_{p_{D_s, D_t}} \left[ \mathrm{R}^{D_t}[Y \mid Z] \right] \tag{70}$$

$$\geq \mathbb{E}_{p_{D_s, D_t}} \left[ \mathrm{R}^{D_t}[Y \mid X] \right] \qquad \text{Lemma 1} \tag{71}$$

$$= \mathrm{R}[Y \mid X] \qquad \text{Assumption 4} \tag{72}$$

We will now show that this lower bound is achieved by an encoder if and only if it satisfies Eqs. (67) and (68), which exist by Prop. 3.

**Sufficiency** ($\Longleftarrow$): Let $p_{Z \mid X}$ be an encoder that satisfies Eqs. (67) and (68). Note that $\mathrm{R}[Y \mid Z] = \mathrm{R}[Y \mid X]$ by Prop. 3. Let $h^* \in \mathcal{H}_Z^*$, then we have the following IDG risk

$$\mathrm{R}_{\mathrm{IDG}}[Y \mid Z] \tag{73}$$

$$= \mathbb{E}_{p_{D_s, D_t}} \left[ \sup_{h \in \mathcal{H}_{Z, D_s}^*} \mathbb{E}_{p_{Z, Y \mid D_t}} [\ell(Y, h(Z))] \right] \tag{74}$$

$$= \mathbb{E}_{p_{D_s, D_t}} \left[ \sup_{h \in \mathcal{H}_{Z, D_s}^*} \mathbb{E}_{p_{Z, Y \mid D_t}} [\ell(Y, h^*(Z))] \right] \qquad \text{Lemma 3 under matching support} \tag{75}$$

$$= \mathbb{E}_{p_{D_t}} \left[ \mathbb{E}_{p_{Z, Y \mid D_t}} [\ell(Y, h^*(Z))] \right] \qquad \text{constant w.r.t } D_s \tag{76}$$

$$= \mathrm{R}[Y \mid Z] = \mathrm{R}[Y \mid X] \tag{77}$$

**Necessity** ($\Longrightarrow$): If the IDG risk is $\mathrm{R}[Y \mid X]$, then it must be the case that

$$\mathrm{R}[Y \mid Z] = \mathrm{R}[Y \mid X] \tag{78}$$

$$\sup_{h \in \mathcal{H}_{Z, d_s}^*} \mathrm{R}_h^{d_t}[Y \mid Z] = \mathrm{R}^{d_t}[Y \mid Z] \quad \forall (d_s, d_t) \in \mathrm{supp}(p_{D_s, D_t}) \tag{79}$$

We will prove by contrapositive that Eq. (79) implies support match (Eq. (68)). Suppose that the support match does not hold. Since $\mathrm{supp}(p_Z) = \cup_{d \in \mathcal{D}} \mathrm{supp}(p_{Z \mid d})$ and $\mathrm{supp}(p_{D_s, D_t}) = \mathcal{D} \times \mathcal{D}$ (Assumption 6), there must exist $(d_s, d_t) \in \mathrm{supp}(p_{D_s, D_t})$ such that $\mathrm{supp}(p_{Z \mid d_s}) \neq \mathrm{supp}(p_{Z \mid d_s})$.

Define the set $S = \mathrm{supp}(p_{Z \mid d_s}) \cap \mathrm{supp}(p_{Z \mid d_t})$ and $\bar{S} = \mathrm{supp}(p_{Z \mid d_t}) \setminus \mathrm{supp}(p_{Z \mid d_s})$, let $\rho = P_{Z \mid d_t}(S)$, and let $h^* \in \mathcal{H}_Z^*$. Then,

$$\sup_{h \in \mathcal{H}_{Z, d_s}^*} \mathrm{R}_h^{d_t}[Y \mid Z] \tag{80}$$

$$= \sup_{h \in \mathcal{H}_{Z, d_s}^*} \rho \, \mathbb{E}_{p_{Y, Z \mid S, d_t}} [\ell(Y, h(Z))] + (1 - \rho) \, \mathbb{E}_{p_{Y, Z \mid \bar{S}, d_t}} [\ell(Y, h(Z))] \tag{81}$$

$$= \sup_{h \in \mathcal{H}_{Z, d_s}^*} \rho \, \mathbb{E}_{p_{Y, Z \mid S, d_t}} [\ell(Y, h^*(Z))] + (1 - \rho) \, \mathbb{E}_{p_{Y, Z \mid \bar{S}, d_t}} [\ell(Y, h(Z))] \qquad \text{Lem. 3} \tag{82}$$

$$= \rho \, \mathbb{E}_{p_{Y, Z \mid S, d_t}} [\ell(Y, h^*(Z))] + (1 - \rho) \sup_{h \in \mathcal{H}_{Z, d_s}^*} \mathbb{E}_{p_{Y, Z \mid \bar{S}, d_t}} [\ell(Y, h(Z))] \tag{83}$$

$$= \mathrm{R}^{d_t}[Y \mid Z] + (1 - \rho) \sup_{h \in \mathcal{H}_{Z, d_s}^*} \mathbb{E}_{p_{Y, Z \mid \bar{S}, d_t}} [\ell(Y, h(Z)) - \ell(Y, h^*(Z))] \tag{84}$$

$$> \mathrm{R}^{d_t}[Y \mid Z] \qquad \text{Lem. 3} \tag{85}$$

Eq. (85) uses the following reasoning. $1 - \rho > 0$ due to support mismatch. For any $h \in \mathcal{H}_{Z, d_s}^*$ such that $h \neq h^*$ on $\bar{S}$ (such an $h$ exists by Item 1 of Assumption 2), we have that

$$\mathbb{E}_{p_{Y, Z \mid \bar{S}, d_t}} [\ell(Y, h(Z)) - \ell(Y, h^*(Z))] > 0 \tag{86}$$

by Lemma 3. $\qquad \square$

As a corollary from the proof strategy we directly have that the optimal DG risk is simply $\mathrm{R}[Y \mid X]$. This means that using the optimal encoder one can actually perform just as well by training on the source as if you were to directly train on the target using the raw data.

**Corollary 2** (Optimal IDG Risk). *Under Assumptions 1 to 6, $\inf_{p_{Z \mid X}} \mathrm{R}_{\mathrm{IDG}}[Y \mid Z] = \mathrm{R}[Y \mid X]$.*

### B.3 Impossibility results

As a direct corollary of Thm. 2 we know that it is impossible to learn an optimal representation without knowledge or assumptions on the target domain. We can actually prove the following much stronger negative result, which essentially states that it is impossible to find a useful representation without having some information about the target domain. Specifically, we prove that if there exists a non-trivial target domain on which the representation is advantageous then there exists an infinite amount of target domains on which it is disadvantageous compared to predicting from a constant.

For clarity, we will focus on the proof for the standard accuracy (0-1 loss) which is much shorter and simpler to understand, but note that we can generalize the proof to all losses with the right assumptions.

The key is that outside of the source domain, the label distribution is unconstrained because generalized covariate shift has no effect. In other words, for any domain which gives some probability mass on an example that has not been seen during training, then all possible labels for that example gives a valid domain. Furthermore, if there exists one domain on which the representation is good, then one can construct a domain on which the representation is bad simply by labelling this point as the constant prediction.

**Proposition 4** (No free lunch for learning representations for IDG, equiv. Proposition 1). *Let $\ell$ be the 0-1 loss with prediction space $\Gamma = \mathcal{Y}$. Let $\mathrm{Rep} : \mathcal{P}(\mathcal{X}, \mathcal{Y}) \to \mathcal{P}(\mathcal{Z}|\mathcal{X})$ be any algorithm for choosing an encoder $p_{Z\,|\,X}$ from the data distribution $p_{X,Y}$, $C$ be any constant r.v. that t.v.i. $\mathcal{Z}$, and $p_{X,Y\,|\,d_s}$ be any desired source distribution such that*

- *there is a unique constant prediction $\gamma_C = \arg\min_{y \in \mathcal{Y}} \mathbb{E}_{p_{Y\,|\,d_s}}[\ell(Y, y)]$,*

- *and $|\mathcal{X} \setminus \mathrm{supp}(p_{X\,|\,d_s})| > 1$.*

*Let $p_{Z_{d_s}\,|\,X} := \mathrm{Rep}(p_{X,Y\,|\,d_s})$ be the chosen source encoder. If there exists a target domain $p_{X,Y\,|\,d_t^g}$ such that*

- *(**Non-trivial support**) $\emptyset \neq \mathrm{supp}(p_{X\,|\,d_t^g}) \subseteq \mathcal{X} \setminus \mathrm{supp}(p_{X\,|\,d_s})$;*

- *(**Satisfies Bayes image invariance**) $\Gamma_{d_t^g}^* = \mathcal{Y}$, i.e., there is at least one example for every possible label;*

- *(**Source encoder is useful**) $p_{Z_{d_s}\,|\,X}$ performs better than a constant representation,*

$$\sup_{h \in \mathcal{H}_{Z_{d_s},d_s}^*} \mathrm{R}_h^{d_t^g}[Y\,|\,Z_{d_s}] < \sup_{h \in \mathcal{H}_{C,d_s}^*} \mathrm{R}_h^{d_t^g}[Y\,|\,C], \tag{87}$$

*Then there exist multiple target domains $d_t^b$ such that $p_{Z_{d_s}\,|\,X}$ underperforms a constant encoder,*

$$\sup_{h \in \mathcal{H}_{Z_{d_s},d_s}^*} \mathrm{R}_h^{d_t^b}[Y\,|\,Z_{d_s}] > \sup_{h \in \mathcal{H}_{C,d_s}^*} \mathrm{R}_h^{d_t^b}[Y\,|\,C]. \tag{88}$$

*Proof.* Let $h^* \in \mathcal{H}_{Z_{d_s},d_s}^*$ be any source Bayes predictor corresponding to our encoder. Partition $\mathcal{Z}$ according to whether $h^*$ predicts like the constant or not:

$$\mathcal{Z}_C := \{z \in \mathcal{Z} \mid h^*(z) = \gamma_C\} \qquad \mathcal{Z}_{\neq C} := \mathcal{Z} \setminus \mathcal{Z}_C. \tag{89}$$

We know by assumption that $d_t^g$ is s.t.

$$\sup_{h \in \mathcal{H}_{Z_{d_s},d_s}^*} \mathrm{R}_h^{d_t^g}[Y\,|\,Z_{d_s}] < \sup_{h \in \mathcal{H}_{C,d_s}^*} \mathrm{R}_h^{d_t^g}[Y\,|\,C], \tag{90}$$

which is clearly only possible if

$$P_{Z_{d_s}\,|\,d_t^g}(\mathcal{Z}_{\neq C}) > 0. \tag{91}$$

In other words, there exists some input $x_{\neq C} \in \mathcal{X} \setminus \mathrm{supp}(p_{X\,|\,d_s})$ that will get represented outside of the constant region, i.e.,

$$P_{Z_{d_s}\,|\,x_{\neq C}}(\mathcal{Z}_{\neq C}) > 0. \tag{92}$$

We will now construct the desired bad domain $d_t^b$ by giving nearly all mass to this $x_{\neq C}$, specifically, let $p_{X \mid d_t^b}(x_{\neq C}) = 1 - \delta$ for some $0 < \delta < 1$. We assign this example to the constant label, i.e., $p_{Y \mid x_{\neq C}, d_t^b}(\gamma_C) = 1$. The rest of the target domain mass $\delta$ is distributed as with the source domain, i.e., $p_{X,Y \mid d_t^b}(x, y) = \delta \cdot p_{X,Y \mid d_s}(x, y)$ for all $x, y \in \operatorname{supp}(p_{X,Y \mid d_s})$. Importantly, the constructed domain $d_t^b$ is valid. Indeed, the Bayes image is the same as the source's (Assumption 5), because we removed no prediction $\gamma$ from the source's Bayes image ($\delta > 0$). We added no new prediction $\gamma$, because $f^*(x_{\neq C}) = \gamma_C \in \mathcal{Y}$ which must already have been in $\Gamma^*$ due to the validity of $d_t^g$.

Now let us compute the desired risk for that "bad" domain and show that the desired encoder performs worse than a constant encoder.

$$\sup_{h \in \mathcal{H}_{Z_{d_s}, d_s}^*} \mathrm{R}_h^{d_t^b}[Y \mid Z_{d_s}] \tag{93}$$

$$= \sup_{h \in \mathcal{H}_{Z_{d_s}, d_s}^*} (1 - \delta) \mathbb{E}_{p_{Z_{d_s} \mid x_{\neq C}}}[1 - \mathbb{1}[\gamma_C = h(Z_{d_s})]] + \delta \, \mathrm{R}_h^{d_s}[Y \mid Z_{d_s}] \tag{94}$$

$$\geq (1 - \delta)(1 - P_{Z_{d_s} \mid x_{\neq C}}(\mathcal{Z}_C)) \tag{95}$$

$$= (1 - \delta)P_{Z_{d_s} \mid x_{\neq C}}(\mathcal{Z}_{\neq C}) \tag{96}$$

In contrast, it is easy to show that $\sup_{h \in \mathcal{H}_{C, d_s}^*} \mathrm{R}_h^{d_t^b}[Y \mid C] \leq \delta$ because the constant predictor would be perfect for $x_{\neq C}$. So any choice of $0 < \delta < \frac{P_{Z_{d_s} \mid x_{\neq C}}(\mathcal{Z}_{\neq C})}{1 + P_{Z_{d_s} \mid x_{\neq C}}(\mathcal{Z}_{\neq C})}$, would satisfy Eq. (88). We conclude the proof by noting that there are infinitely many such choices of $\delta$, and any choice of those would result in a different valid bad domain $d_t^b$.

$\square$

Note that representations can often be much worse than using a constant r.v. Specifically, if an encoder $p_{Z \mid X}$ maps an $x$ outside of the source support then there exists an infinite number of target domains where that representation is the worst possible representation.

**Proposition 5** (Worst representation). *Let* $\mathrm{Rep}, p_{Y, X \mid d_s}, p_{Z_{d_s} \mid X}, \ell$ *be as in Prop. 4, and* $\epsilon > 0$. *If there exists an example* $x_b \in \mathcal{X} \setminus \operatorname{supp}(p_{X \mid d_s})$ *that is mapped outside of the source support, i.e.,* $\operatorname{supp}(p_{Z_{d_s} \mid x_b}) \cap \operatorname{supp}(p_{Z \mid d_s}) = \emptyset$, *then there exist many target domains* $p_{X,Y \mid d_t}$ *s.t.* $p_{Z_{d_s} \mid X}$ *is* $\epsilon$ *close to the worst possible loss, i.e.,*

$$\sup_{h \in \mathcal{H}_{Z_{d_s}, d_s}^*} \mathrm{R}_h^{d_t}[Y \mid Z_{d_s}] \geq 1 - \epsilon. \tag{97}$$

*Proof.* By assumption there exists an $x_b$ whose support is outside the source support. Then similarly to Prop. 4 we construct a bad target domain $d_t$ by giving nearly all mass to that example $p_{X \mid d_t}(x_b) = 1 - \delta$ where $\delta > 0$ and assign with probability 1 to some label that is in the source Bayes image, i.e., $p_{Y \mid x_b, d_t}(\gamma_b) = 1$ for some $\gamma_b \in \Gamma_{d_s}^*$. The rest of the target domain mass $\delta$ is distributed as in Prop. 4 to the source inputs. As in Prop. 4, such a target domain $d_t$ satisfies our assumptions. Now let us compute the risk for that $d_t$ and show that the desired encoder performs arbitrarily bad.

$$\sup_{h \in \mathcal{H}_{Z_{d_s}, d_s}^*} \mathrm{R}_h^{d_t}[Y \mid Z_{d_s}] \tag{98}$$

$$= \sup_{h \in \mathcal{H}_{Z_{d_s}, d_s}^*} (1 - \delta) \mathbb{E}_{p_{Z_{d_s} \mid x_b}}[1 - \mathbb{1}[\gamma_b = h(Z_{d_s})]] + \delta \, \mathrm{R}_h^{d_s}[Y \mid Z_{d_s}] \qquad \text{Eq. (94)} \tag{99}$$

$$\geq \sup_{h \in \mathcal{H}_{Z_{d_s}, d_s}^*} (1 - \delta) \mathbb{E}_{p_{Z_{d_s} \mid x_b}}[1 - \mathbb{1}[\gamma_b = h(Z_{d_s})]] \tag{100}$$

$$= 1 - \delta \tag{101}$$

Eq. (101) uses the fact that $\mathcal{H}_{Z_{d_s}, d_s}^*$ is unconstrained outside of the source support and that by assumption $\operatorname{supp}(p_{Z_{d_s} \mid x_b}) \cap \operatorname{supp}(p_{Z_{d_s} \mid d_s}) = \emptyset$. To achieve the sup $1 - \delta$ it then suffices to predict an $\gamma \neq \gamma_b \in \Gamma$. We thus see that Eq. (97) holds for $d_t$ as long as $0 < \delta < \epsilon$. We conclude the proof by noting that there is an infinite possible choices of $\delta$ each of which give rise to a bad target domain.

$\square$

### B.4 Augmentations

Proposition 2 shows that the optimal representations for IDG can be learned with augmentations in a self-supervised fashion. Here, we provide formal definitions, assumptions, and proofs.

**Definition 6** (Augmenter). An *augmenter* is a conditional distribution $p_{A \mid X} : \mathcal{A} \times \mathcal{X} \to [0, 1]$ from the input space $\mathcal{X}$ to an augmentation space $\mathcal{A}$. For example, in CLIP $\mathcal{X}$ is the space of images and $\mathcal{A}$ is the space of text. In standard SSL, $\mathcal{A}$ is typically the same as $\mathcal{X}$ (e.g., both $\mathcal{X}$ and $\mathcal{A}$ are the space of images).

**Definition 7** (Augmentation conditional set). Given an augmenter $p_{A \mid X}$, define the augmentation conditional set as the set of conditionals of $A$ given $X$:

$$\mathcal{P}^*(A \mid X) := \left\{ p_{A \mid x} \mid x \in \mathcal{X} \right\} \tag{102}$$

Similarly, we can define the augmentation conditional set for domain $d$:

$$\mathcal{P}_d^*(A \mid X) := \left\{ p_{A \mid x} \mid x \in \operatorname{supp}(p_{X \mid d}) \right\} \tag{103}$$

These sets are clearly countable. Note that the augmentation conditional set can be seen as a special case of the Bayes image (Def. 3) if we view the augmentation $A$ as the label and consider the log-loss where the conditional distribution is the Bayes optimal predictor due to its strict properness (Gneiting & Raftery, 2007).

**Assumption 7** (Finite augmentation entropy). We consider the augmenter $p_{A \mid X}$ such that the entropy of the augmentation $A$ is finite, i.e., $\mathrm{H}[A] < \infty$.

**Assumption 8** (Cardinalities). We assume that

$$|\mathcal{Z}| \geq |\mathcal{P}^*(A \mid X)| \tag{104}$$

This is a similar assumption as Assumption 3, which ensures the existence of optimal representations.

**Assumption 9** (Domain-agnostic augmentation). We assume that the augmentation $A$ is *domain-agnostic*, i.e., the augmentation conditional set is invariant across domains,

$$\mathcal{P}_d^*(A \mid X) = \mathcal{P}^*(A \mid X), \quad \forall d \in \mathcal{D} \tag{105}$$

This assumption is generalized from the constant Bayes image assumption (Assumption 5), which guarantees the existence of optimal representations.

Domain-agnostic augmentations essentially ensures that each augmentation conditional $p_{A \mid x} \in \mathcal{P}^*(A \mid X)$ is seen at least once in all domains. If we introduce an equivalence relation $\sim$ as $x \sim x'$ iff $p_{A \mid x} = p_{A \mid x'}$ and the equivalence class $[x] := \{x' \in \mathcal{X} \mid x' \sim x\}$. Under this relation, it is easy to see that the above assumption is satisfied if and only if, for all possible equivalence classes $[x] \in \{[x'] \mid x' \in \mathcal{X}\}$, we have that $[x]$ has intersections with all domains:

$$[x] \cap \operatorname{supp}(p_{X \mid d}) \neq \emptyset, \quad \forall d \in \mathcal{D} \tag{106}$$

Not all augmentations are domain-agnostic. In particular, the standard image augmentations used by typical SSL models like SimCLR are not domain-agnostic, but the text-image augmentations of CLIP nearly are, as discussed in the main body (Sec. 4).

**Assumption 10** (Bayes-preserving augmentation). We assume that the augmentation $A$ is *Bayes-preserving*, i.e., $\forall x, x' \in \mathcal{X}$,

$$p_{A \mid x} = p_{A \mid x'} \implies f^*(x) = f^*(x'). \tag{107}$$

Under the notion of equivalence relation in Assumption 9, this means that for each equivalence class $[x]$, all $x' \in [x]$ have the same Bayes prediction. Note that most augmentations used in practice like standard image augmentations are Bayes-preserving.

Next, we show that under the above assumptions, we can learn optimal representations by maximizing the mutual information $\mathrm{I}[A; Z]$ (in the case of log-loss $\ell$) under the support match constraint. We use log-loss simply because it is typically the loss used for training in practice. Note that the learned representations are optimal for any strict proper losses.

**Proposition 6** (Learning optimal representations without labels, equiv. Proposition 2). *Let $p_{A \mid X}$ be an augmenter. Under Assumptions 1 to 10, any encoder $p_{Z \mid X}$ such that*

$$p_{Z \mid X} \in \arg \max_{p_{Z \mid X}} \ I[A; Z] \tag{108}$$

$$\text{s.t.} \quad \forall \, d \in \mathcal{D}, \ \text{supp}(p_{Z \mid d}) = \text{supp}(p_Z) \tag{109}$$

*is optimal for idealized domain generalization.*

*Proof.* The support match constraint Eq. (109) is equivalent to the support match constraint Eq. (68). Thus, Prop. 3 and Thm. 2 state that we only need to prove that maximizing the mutual information of $A$ and $Z$ under the support constraint implies that

$$\text{R}[Y \mid Z] = \text{R}[Y \mid X]. \tag{110}$$

We will prove this by constructing an optimal predictor $h^*$.

Since $\text{H}[A] < \infty$ (Assumption 7) we have that

$$\arg \max_{p_{Z \mid X}} I[A; Z] = \arg \min_{p_{Z \mid X}} \text{H}[A \mid Z]. \tag{111}$$

Note the fact that the conditional entropy is the Bayes risk under the log-loss (Gneiting & Raftery, 2007), i.e., $\text{H}[A \mid Z] = \text{R}[A \mid Z]$. By construction, $A$ satisfies covariate shift w.r.t. $X$ (thus Bayes invariant) since $A - X - D$ forms a Markov chain. Together with Assumptions 1 and 7 to 9, it means that the optimization problem in Eqs. (108) and (109) satisfies the assumptions of Prop. 3, with $A$ in place of $Y$. Thus, an optimal encoder satisfies $\text{R}[A \mid Z] = \text{R}[A \mid X]$, which leads to

$$\text{H}[A \mid Z] = \text{H}[A \mid X]. \tag{112}$$

By Assumption 7, we can invoke Lemma 2 with the fact that $A - X - Z$ forms a Markove chain to show that for all $(x, z) \in \text{supp}(p_{X,Z})$

$$p_{A \mid z} = p_{A \mid x}, \tag{113}$$

as the conditional distributions are the Bayes optimal predictors due to strict properness of log-loss.

Now, define the following equivalence relation on $\mathcal{X}$,

$$x \sim x' \quad \iff \quad p_{A \mid x} = p_{A \mid x'}. \tag{114}$$

Because the number of equivalence classes under $\sim$ is countable, there exists a maximal invariant $M : \mathcal{X} \to \mathbb{N}$ from $\mathcal{X}$ to the natural numbers (for our definition of a maximal invariant see Definition 2, Dubois et al., 2021). By Assumption 10, $f^*$ is invariant on the equivalence classes $[x] := \{x' \in \mathcal{X} \mid x' \sim x\}$ for all $x \in \mathcal{X}$. Thus, there exists a function $g : \mathbb{N} \to \mathcal{A}$ such that $f^* = g \circ M$ (Lemma 5, Dubois et al., 2021). Given $z \in \text{supp}(p_Z)$, we construct $h^*$ in the following way. Let $x_z \in \text{supp}(p_{X \mid z})$ be any input point that could have led to this representation $z$ and define

$$h^*(z) = g(M(x_z)). \tag{115}$$

By Eq. (113) we are guaranteed that all $x \in \text{supp}(p_{X \mid z})$ share the same value for $f^*$ since they are in the same equivalence class. Thus, by the definition of $M$ we have that

$$M(x_Z) = M(X) \quad \text{for} \quad (X, Z) \sim p_{X,Z}. \tag{116}$$

Therefore,

$$\text{R}_{h^*}[Y \mid Z] = \mathbb{E}_{p_{Y,Z}}[\ell(Y, h^*(Z)] \tag{117}$$

$$= \mathbb{E}_{p_{Y \mid X} p_{X,Z}}[\ell(Y, h^*(Z)] \tag{118}$$

$$= \mathbb{E}_{p_{Y \mid X} p_{X,Z}}[\ell(Y, g(M(x_Z)))] \qquad \text{Eq. (115)} \tag{119}$$

$$= \mathbb{E}_{p_{Y \mid X} p_{X,Z}}[\ell(Y, g(M(X)))] \qquad \text{Eq. (116)} \tag{120}$$

$$= \mathbb{E}_{p_{Y \mid X} p_{X,Z}}[\ell(Y, f^*(X))] = \text{R}[Y \mid X]. \tag{121}$$

$\square$

## C  PRACTICAL OBJECTIVES

Proposition 6 provides an objective to obtain the desired optimal representations, compared to Thm. 2 it is more practical in that it does not require direct access to the labels and in that it can use augmentations under appropriate assumptions. There are nevertheless multiple remaining issues for deriving objectives that can be trained with in practice. Specifically, (i) the support constraint is hard to satisfy in practice; (ii) mutual information $I[A; Z]$ is hard to estimate from samples (Poole et al., 2019); (iii) the objective is constrained which is harder to optimize. We will now show different objectives and variational bounds of them that do not suffer from these issues, and could still recover the desired encoders in their optima. In contrast to the proofs of main theoretical results (previous section), here the derivations will be less formal.

As we have seen in Proposition 6, the optimal representation achieves $I[A; Z] = I[A; X]$. In the following, we will rewrite the objective as the constrained optimization:

$$p_{Z\,|\,X} \in \arg \min_{p_{Z\,|\,X}} B[Z, X, Y, D] \tag{122}$$

$$\text{s.t.} \quad I[A; Z] = I[A; X] \tag{123}$$

where we introduce the *domain bottleneck* $B[Z, X, Y, D]$ as the objective for enforcing support match (which we denote as $B[Z, D]$ in the main body for simplicity). The requirement on the domain bottleneck objective is that minimizing Eq. (122) under Eq. (123) implies that the support match constraint holds (and can be achieved by some encoder), which leads to optimal representations for IDG. Different domain bottlenecks will be derived later this section. We can then use Lagrangian relaxation to get the following unconstrained objectives.

$$\arg \min_{p_{Z\,|\,X}} \quad -I[A; Z] + \lambda\, B[Z, X, Y, D] \tag{124}$$

The first term can be easily optimized using variational bounds on MI. Throughout the paper, we will use a contrastive variational lower bound which is based on InfoNCE (Oord et al., 2018). Namely, let $X$ be the input sample and $A$ be the 'positive' augmentation sampled from $p_{A\,|\,X}$. We then obtain $n$ 'negative' augmentations $\{A_i^-\}_{i=1}^n$ by first independently sampling $\{X_i^-\}_{i=1}^n$ from the marginal $p_X$ and then sampling $A_i^-$ from $p_{A\,|\,X_i^-}$. It is easy to see that the negatives $A_i^-$ follow the marginal $p_A$. We construct $\mathbf{A} := \{A, A_1^-, \ldots, A_n^-\}$. Let $Z$ be the representation of $X$ by passing it through the encoder $p_\varphi := p_{Z\,|\,X}$ parameterized by $\varphi$ and $s_\psi$ the critic function parametrized by $\psi$ used to score which $A' \in \mathbf{A}$ is the positive augmentation. Then we have the following variational lower bound (Poole et al., 2019):

$$I[A; Z] \geq \log(n + 1) + \mathbb{E}_{p_{\mathbf{A}, X, Z}} \left[ \log \frac{\exp s_\psi(A, Z)}{\sum_{A' \in \mathbf{A}} \exp s_\psi(A', Z)} \right] \tag{125}$$

In the case of unconstrained variational families $s_\psi, p_\varphi$ and infinite samples ($n \to \infty$), the above variational bound recovers $I[A; Z]$ up to a constant (see Oord et al. (2018); Dubois et al. (2021)). Typically the critic is separable, i.e., $s_\psi(A, Z) := g_\psi(A)^T h_\psi(Z)$. As discussed in the main body, it can be tied with the encoder $p_\varphi$ when $\mathcal{A} = \mathcal{X}$.

In the following we focus on the second term $B[Z, X, Y, D]$ and discuss several choices.

Throughout this section, the function $M : \mathcal{X} \to \mathbb{N}$ is the maximal invariant defined in Prop. 6 via the equivalence relation defined in Eq. (114).

### C.1  MUTUAL INFORMATION BOTTLENECK $B[Z, X, Y, D] = I[Z; X]$

The first bottleneck we consider is so called mutual information (MI) bottleneck $B[Z, X, Y, D] = I[Z; X]$, which was introduced by Tishby et al. (2000) to achieve a tradeoff between the predictive power and the complexity of representations. Intuitively, it tries to remove all information of $Z$ that is not needed for maximizing $I[Z; A]$. In particular, using the fact that $Z - X - D$ forms a Markov chain and the chain rule of MI, we have $I[Z; X] = I[Z; X, D] = I[Z; D] + I[Z; X \mid D]$. Thus, it not only minimizes $I[Z; D]$, i.e., matches the representations' distribution *across* domains, but also minimizes $I[Z; X \mid D]$, i.e., matches the representations' distribution *inside* domains.

**Why** The key to show is that minimizing Eq. (122), i.e., $\arg\min_{p_{Z\,|\,X}} I[Z;X]$ under $I[A;Z] = I[A;X]$, implies the support match constraint. This can be seen as a specific subcase of Dubois et al.'s (2021) Corollary 15 with $A$ in place of $Y$ and $M(X)$ induced by $p_{A\,|\,X}$ as in the proof of Prop. 6. From the corollary, we know that $\min_{p_{Z\,|\,X}} I[Z;X] = H[M(X)]$ which can be achieved by any $Z$ s.t. $p_{Z\,|\,x} = p_{Z\,|\,x'} \iff M(x) = M(x')$. With the assumption of domain-agnostic augmentations (Assumption 9), we have that the set of maximal invariant $\{M(x)\,|\,x \in \text{supp}(p_{X\,|\,d})\}$ is invariant across domains. Then we directly have $\text{supp}(p_{Z\,|\,d}) = \cup_{x\in\text{supp}(p_{X\,|\,d})}\text{supp}(p_{Z\,|\,x}) = \cup_{x\in\text{supp}(p_X)}\text{supp}(p_{Z\,|\,x}) = \text{supp}(p_Z)$, where we use the fact that $x$ within the same equivalence class has the the same $p_{Z\,|\,x}$.

**How** Essentially, we can use any variational upper bound of mutual information. We consider the one used by Variational Information Bottelenck (Alemi et al., 2016), i.e.,

$$I[Z;X] = \mathbb{E}_{p_{X,Z}}\left[\log \frac{p_\varphi(Z\,|\,X)}{p_Z(Z)}\right] \tag{126}$$

$$= \mathbb{E}_{p_{X,Z}}\left[\log \frac{p_\varphi(Z\,|\,X)}{q_\theta(Z)}\right] - D_{\text{KL}}[p_Z(Z)\|q_\theta(Z)] \tag{127}$$

$$\leq \mathbb{E}_{p_{X,Z}}\left[\log \frac{p_\varphi(Z\,|\,X)}{q_\theta(Z)}\right] \tag{128}$$

$$= \mathbb{E}_{p_X}[D_{\text{KL}}[p_\varphi(Z\,|\,X)\|q_\theta(Z)]] \tag{129}$$

where a variational distribution $q_\theta$ is used to approximate $p_Z$ and is jointly optimized with $p_\varphi$ to minimize the bound. The approximation gap of the bound is $D_{\text{KL}}[p_Z(Z)\|q_\theta(Z)]$. Ignoring the constant, the final loss becomes

$$\mathcal{L}_{\text{MI}}(\psi, \varphi, \theta) := \mathbb{E}_{p_{X,\mathbf{A},Z}}\left[-\log \frac{\exp s_\psi(A, Z)}{\sum_{A'\in\mathbf{A}} \exp s_\psi(A', Z)} + \lambda\, D_{\text{KL}}[p_\varphi(Z\,|\,X)\|q_\theta(Z)]\right] \tag{130}$$

which recovers the optimal encoder in the case of unconstrained variational families for $p_\varphi, q_\theta, s_\psi$, infinite samples $n \to \infty$, and any $\lambda > 1$ (Dubois et al., 2021).

## C.2 Entropy bottleneck $B[Z, X, Y, D] = H[Z]$

The entropy (Ent) bottleneck introduced in the main body is a special case of the MI bottleneck, where the encoder is a deterministic mapping, i.e., $p_\varphi(Z\,|\,x)$ is a dirac delta function for all $x \in \mathcal{X}$ and we denote by $e_\varphi(x)$ the deterministic encoder s.t. $p_\varphi(e_\varphi(x)\,|\,x) = 1$.

**Why** In the deterministic case, the MI bottleneck becomes the entropy bottleneck because $I[X;Z] = H[Z] - H[Z\,|\,X] = H[Z]$, where we use the fact that $H[Z\,|\,X] = 0$. Importantly, considering only deterministic encoders does not constrain our ability to learning optimal encoders. Indeed, just as with the MI bottleneck optimizing the objective with the entropy bottleneck under $I[A;Z] = I[A;X]$ will recover encoders s.t. $e_\varphi(x) = e_\varphi(x') \iff M(x) = M(x')$, which also satisfies the support match constraint as discussed before.

**How** Using the same derivation as the MI bottleneck, we can derive the variational upper bound on entropy

$$H[Z] \leq \mathbb{E}_{p_Z}[-\log q_\theta(Z)] \tag{131}$$

which is the standard variational bound used in neural compression (Ballé et al., 2016; Theis et al., 2017). Putting all together, we have

$$\mathcal{L}_{\text{Ent}}(\psi, \theta, \varphi) := \mathbb{E}_{p_{X,\mathbf{A},Z}}\left[-\log \frac{\exp s_\psi(A, Z)}{\sum_{A'\in\mathbf{A}} \exp s_\psi(A', Z)} - \lambda \log q_\theta(Z)\right] \tag{132}$$

which also recovers the optimal encoder with unconstrained variational families, infinite samples, and $\lambda > 1$ as with the MI bottleneck. The detailed algorithm is provided in Algorithm 2. Note that the discreteness of $Z$ could lead to difficulty of gradient-based optimization, and we follow Ballé et al. (2016) to add uniform noise to $Z$ as a differentiable substitute for rounding during training. In our experiments, we will mostly use the Ent bottleneck instead of the MI bottleneck to avoid introducing stochastic encoders.

---

**Algorithm 2** Ent objective

---

**Require:** $e_\varphi, s_\psi, q_\theta, X, n$
 1: $Z \leftarrow e_\varphi(X)$
 2: $A \leftarrow \text{sample}(p_{A\,|\,X})$
 3: $\{(X_i^-, A_i^-)\}_{i=1}^n \xleftarrow{\text{i.i.d.}} \text{sample}(p_{X,A})$
 4: $\mathbf{A} \leftarrow \{A\} \cup \{A_i^-\}_{i=1}^n$
 5: $\mathcal{L}_{\text{aug}} \leftarrow -\log \frac{\exp s_\psi(A,Z)}{\sum_{A' \in \mathbf{A}} \exp s_\psi(A',Z)}$  $\quad \triangleright -\text{I}[A; Z]$
 6: $\mathcal{L}_{\text{supp}} \leftarrow -\log q_\theta(Z)$  $\qquad\qquad\qquad \triangleright \text{H}[Z]$
 7: **return** $\mathcal{L}_{\text{Ent}} = \mathcal{L}_{\text{aug}} + \lambda \mathcal{L}_{\text{supp}}$

---

## C.3 CONTRASTIVE ADVERSARIAL DOMAIN BOTTLENECK $\text{B}[Z, X, Y, D] = \text{I}[Z; D]$

The previous two bottlenecks require removing the information of $Z$ (about $X$) as much as possible, which seems to be unnecessary since our ultimate goal is to match the support of $Z$ across domains. Now we introduce a bottleneck $\text{B}[Z, X, Y, D] = \text{I}[Z; D]$ which we only seek to remove the information of $Z$ about the domain $D$. This is very related to the work on invariant representation learning for domain generalization/adaptation (e.g., Ganin et al., 2016; Li et al., 2018a). We derive a new variational bound called the contrastive adversarial domain (CAD) bottleneck that is more stable to train and leads to better empirical performance. For simplicity we consider the deterministic encoder $e_\varphi(x)$ as with the main body.

**Why**  Similar to the previous analysis, we aim to show that $\arg\min_{p_{Z\,|\,X}} \text{I}[Z; D]$ under $\text{I}[A; Z] = \text{I}[A; X]$ leads to the support match constraint. Using Eq. (116) we have $\text{I}[Z; D] = \text{I}[Z, M(X_Z); D] = \text{I}[Z, M(X); D] = \text{I}[M(X); D] + \text{I}[Z; D \,|\, M(X)]$ where the last equality uses the chain rule of mutual information. Due to the non-negativity of (conditional) mutual information, we have that the minimum of $\text{I}[Z; D]$ under $\text{I}[A; Z] = \text{I}[A; X]$ is $\text{I}[M(X); D]$. Then we show the minimum is achievable by constructing the same optimal encoder $e_\varphi(X)$ as the Ent bottleneck which clearly satisfies $\text{I}[Z; D \,|\, M(X)] = 0$. It is then easy to show that the support match constraint has to hold when $\text{I}[Z; D \,|\, M(X)] = 0$ by contrapositive. Indeed, suppose that the support constraint does not hold then it must be true that $\text{I}[Z; D \,|\, M(X)] > 0$ and so the encoder cannot be optimal.

**How**  The typical way of minimizing $\text{I}[Z; D]$ is to derive the variational bound as

$$\text{I}[Z; D] = \text{H}[D] - \text{H}[D \,|\, Z] \tag{133}$$

$$= (const) - \mathbb{E}_{p_{D,z}}\big[-\log p_{D\,|\,Z}(D \,|\, Z)\big] \tag{134}$$

$$\geq (const) - \mathbb{E}_{p_{D,z}}\big[-\log q_\phi(D \,|\, Z)\big] \tag{135}$$

where a variational distribution (or domain classifier) $q_\phi$ is used to approximate $p_{D\,|\,Z}$ and jointly trained to *maximize* the bound. This recovers the domain-adversarial training method as introduced in Ganin et al. (2016). However, this has two potential issues: 1) it gives a *lower* bound instead of the desired upper bound on $\text{I}[Z; D]$; 2) it requires adversarial training which is not stable in practice (Goodfellow, 2016; Kodali et al., 2017).

We propose the contrastive adversarial domain (CAD) bottleneck, which is based on the above explicit version but uses a variational distribution $q_\phi(D \,|\, Z)$ that is *tied with other parts of the model*, thus no need to learn a domain classifier. Suppose we have access to a set of inputs $\mathbf{X}$, we first introduce a contrastive variational distribution $q_{\varphi,\mathbf{x}}(X \,|\, Z)$ of $p_{X\,|\,Z}$ as

$$q_{\varphi,\mathbf{x}}(X \,|\, Z) := \frac{\exp s_\varphi(X, Z)}{\sum_{X' \in \mathbf{X}} \exp s_\varphi(X', Z)} \tag{136}$$

where $s_\varphi(X, Z) := e_\varphi(X)^T Z$ is tied with the encoder $e_\varphi$. Note that $q_{\varphi,\mathbf{x}}$ has support over $\mathbf{X}$, and equals $p_{X\,|\,Z}$ when $s_\varphi(X, Z) \propto \log p_{X,Z}(X, Z)$ and $\mathbf{X}$ recovers $\mathcal{X}$. In practice, we use a variety of crude approximations. Our first crude approximation is that we use the minibatch of samples, i.e., the independently sampled $\{X_i^-\}_{i=1}^n$ as $\mathbf{X}$. Now, since $p_{D\,|\,Z}$ can be rewritten as $\mathbb{E}_{p_{X\,|\,Z}}\big[p_{D\,|\,X}\big]$ using the fact that $D - X - Z$ forms a Markov chain, we obtain the following variational distribution:

$$q_{\varphi,\mathbf{x}}(D \,|\, Z) = \mathbb{E}_{q_{\varphi,\mathbf{x}}}\big[p_{D\,|\,X}(D \,|\, X)\big] \tag{137}$$

which recovers $p_{D\,|\,Z}$ when $q_{\varphi,\mathbf{x}} = p_{X\,|\,Z}$. Note that $p_{D\,|\,X}$ is still not available. For our second crude approximation, we use a count estimate $\hat{p}_{\mathbf{D},\mathbf{X}}$. In particular, we obtain a collection $\mathbf{D}$ by taking each $X' \in \mathbf{X}$ and independently sampling $D'$ from $p_{D\,|\,X'}$ to get $\mathbf{D} := \{D_i^-\}_{i=1}^n$. In other words, $\{(D_i^-, X_i^-)\}_{i=1}^n$ are all i.i.d. sampled from $p_{D,X}$. Then we use a count estimate

$$\hat{p}_{\mathbf{D},\mathbf{X}}(d\,|\,x) = \frac{\sum_{i=1}^n \mathbb{I}\,(X_i^- = x, D_i^- = d)}{\sum_{i=1}^n \mathbb{I}\,(X_i^- = x)} \tag{138}$$

which is an accurate estimate with infinite samples. This leads to our final variational distribution:

$$q_{\varphi,\mathbf{x},\mathbf{D}}(D\,|\,Z) = \sum_{X' \in \mathbf{X}} q_{\varphi,\mathbf{x}}(X'\,|\,Z)\hat{p}_{\mathbf{D},\mathbf{X}}(D\,|\,X') \tag{139}$$

Putting all together we get that the loss:

$$\mathcal{L}_{\text{CAD}}(\varphi, \psi) := \mathbb{E}_{p_{\mathbf{D},\mathbf{X},\mathbf{A},Z}} \left[ -\log \frac{\exp s_\psi(A, Z)}{\sum_{A' \in \mathbf{A}} \exp s_\psi(A', Z)} + \lambda \log \left( \sum_{X' \in \mathbf{X}} q_{\varphi,\mathbf{x}}(X'\,|\,Z)\hat{p}_{\mathbf{D},\mathbf{X}}(D\,|\,X') \right) \right].$$
$$\tag{140}$$

In practice, $\hat{p}_{\mathbf{D},\mathbf{X}}(D\,|\,X)$ is typically a dirac delta function since it is rare to have the same samples in a minibatch. Thus, in Eq. (139) we only need to sum $q_{\varphi,\mathbf{x}}(X'\,|\,Z)$ over those associated with the same domain label $D$ as $X$, i.e., $\mathbf{X}_D := \{X_i^- \,|\, D_i^- = D, i \in [n]\}$ where $[n] := \{1, \ldots, n\}$. This leads to the simplified loss:

$$\mathcal{L}_{\text{CAD}}(\varphi, \psi) := \mathbb{E}_{p_{\mathbf{D},\mathbf{X},\mathbf{A},z}} \left[ -\log \frac{\exp s_\psi(A, Z)}{\sum_{A' \in \mathbf{A}} \exp s_\psi(A', Z)} + \lambda \log \left( \sum_{X' \in \mathbf{X}_D} q_{\varphi,\mathbf{x}}(X'\,|\,Z) \right) \right]. \tag{141}$$

In practice, we find that the second term that minimizes the log probability leads to numerical instability. Intuitively, this could be seen by the exploding gradient of the function $\log(p)$ when $p \to 0$. We thus replace it with $-\log(1 - p)$ which has the same optima. I.e. in practice we maximize the log of the probablity summed over $\mathbf{X}_{\neg D} := \mathbf{X} \setminus \mathbf{X}_D$. This reduces Eq. (141) to Eq. (7) described in the main body with a detailed algorithm in Algorithm 1. Note that it is easy to generalize Algorithm 1 to parallel computation within a batch of samples. Indeed, for *each* sample in the batch, we can view all other samples in the batch as negatives and compute the loss efficiently in parallel.

## C.4 CONDITIONAL CAD $\text{B}[Z, X, Y, D] = \text{I}[Z; D\,|\,Y]$

The analysis of the CAD bottleneck also implies that we can minimize the conditional mutual information $\text{I}[Z; D\,|\,M(X)]$ if we have access to $M(X)$. However, since $M(X)$ is typically not available in practice, we consider the special case where $M(X) = Y$. In particular, this is the case where the labels are available and the supervised augmentations are used (see Fig. 2b). This reduces the bottleneck to $\text{B}[Z, X, Y, D] = \text{I}[Z; D\,|\,Y]$ which is related to the conditional version of the domain-adversarial neural network (Li et al., 2018b). In practice, minimizing $\text{I}[Z; D\,|\,Y]$ could be easier for optimization than $\text{I}[Z; D]$, as it does not require to remove the information that $D$ has about $Y$. In the following, we derive the conditional CAD ($\text{C}^2\text{AD}$) bottleneck using a similar idea as CAD.

**How** In this case, we want to minimize

$$\text{I}[Z; D\,|\,Y] = \text{H}[D\,|\,Y] - \text{H}[D\,|\,Z, Y] \tag{142}$$
$$= (const) - \text{H}[D\,|\,Z, Y] \tag{143}$$
$$\geq (const) - \mathbb{E}_{p_{D,Z,Y}}[-\log q(D\,|\,Z, Y)] \tag{144}$$

where $q(D\,|\,Z, Y)$ is a variational distribution of $p_{D\,|\,Z,Y}$. Similar to the unconditional case, we also aim to use a non-parametric approximation that is tied with other parts of the model, and we obtain it using the fact $p_{D\,|\,Z,Y} = \mathbb{E}_{p_{X\,|\,Z,Y}}[p_{D\,|\,X}]$. Specifically, let $Y$ be the label of input $X$ sampled from $p_{Y\,|\,X}$ and $\mathbf{Y} := \{Y_1^-, \ldots, Y_n^-\}$ be the collection of labels obtained by independently sampling the label from $p_{Y\,|\,X'}$ for each $X' \in \mathbf{X}$. We collect samples associated with the label $Y$, i.e., $\mathbf{X}_Y := \{X_i^- \,|\, Y_i^- = Y, i \in [n]\}$ and obtain a variational distribution of $p_{X\,|\,Z,Y}$:

$$q_{\varphi,\mathbf{x},\mathbf{Y}}(X\,|\,Z, Y) := \frac{\exp s_\varphi(X, Z)}{\sum_{X' \in \mathbf{X}_Y} \exp s_\varphi(X', Z)} \tag{145}$$

where we use the same critic $s_\varphi(X, Z) := e_\varphi(X)^T Z$ that is tied with the encoder $e_\varphi$ as before, but only take softmax over those samples with the same label $Y$. For the term $p_{D\,|\,X}$, we use the same count estimate $\hat{p}_{\mathbf{D},\mathbf{X}}$ in Eq. (138). Then we obtain the variational distribution of $p_{D\,|\,Z,Y}$:

$$q_{\varphi,\mathbf{X},\mathbf{D},\mathbf{Y}}(D\,|\,Z,Y) = \sum_{X' \in \mathbf{X}_Y} q_{\varphi,\mathbf{X},\mathbf{Y}}(X'\,|\,Z,Y)\hat{p}_{\mathbf{D},\mathbf{X}}(D\,|\,X') \tag{146}$$

Putting all together we get that the final loss:

$$\mathcal{L}_{\mathrm{C}^2\mathrm{AD}}(\varphi, \psi) := \mathbb{E}_{p_{\mathbf{D},\mathbf{X},\mathbf{A},\mathbf{Y},Z}}\left[ -\log \frac{\exp s_\psi(A, Z)}{\sum_{A' \in \mathbf{A}} \exp s_\psi(A', Z)} + \lambda \log\left( \sum_{X' \in \mathbf{X}_Y} q_{\varphi,\mathbf{X},\mathbf{Y}}(X'\,|\,Z,Y)\hat{p}_{\mathbf{D},\mathbf{X}}(D\,|\,X') \right) \right]. \tag{147}$$

Again, since in practice $\hat{p}_{\mathbf{D},\mathbf{X}}(D\,|\,X)$ is typically a dirac delta function, the summation in Eq. (146) can be done only over those associated with the same label $Y$ *and* the same domain label $D$ as $X$, i.e., $\mathbf{X}_{Y,D} := \{X_i^- \,|\, Y_i^- = Y, D_i^- = D, i \in [n]\}$. Similarly, instead of minimizing the log of the probability summed over $\mathbf{X}_{Y,D}$, we maximize the log of the probability summed over $\mathbf{X}_{Y,\neg D} := \mathbf{X}_Y \setminus \mathbf{X}_{Y,D} = \{X_i^- \,|\, Y_i^- = Y, D_i^- \neq D, i \in [n]\}$. Finally we obtaine the simplified loss:

$$\mathcal{L}_{\mathrm{C}^2\mathrm{AD}}(\varphi, \psi) := \mathbb{E}_{p_{\mathbf{D},\mathbf{X},\mathbf{A},\mathbf{Y},Z}}\left[ -\log \frac{\exp s_\psi(A, Z)}{\sum_{A' \in \mathbf{A}} \exp s_\psi(A', Z)} - \lambda \log\left( \sum_{X' \in \mathbf{X}_{Y,\neg D}} q_{\varphi,\mathbf{X},\mathbf{Y}}(X'\,|\,Z,Y) \right) \right]. \tag{148}$$

A detailed algorithm is in Algorithm 3.

---

**Algorithm 3** conditional CAD ($\mathrm{C}^2\mathrm{AD}$) objective

---

**Require:** $e_\varphi, s_\psi, D, X, Y, n$
1: $Z \leftarrow e_\varphi(X)$
2: $A \leftarrow \mathrm{sample}(p_{A\,|\,X})$
3: $\left\{(D_i^-, X_i^-, A_i^-, Y_i^-)\right\}_{i=1}^n \xleftarrow{\text{i.i.d.}} \mathrm{sample}(p_{D,X,A,Y})$
4: $\mathbf{X}, \mathbf{A} \leftarrow \left\{X_i^-\right\}_{i=1}^n, \{A\} \cup \left\{A_i^-\right\}_{i=1}^n$
5: $\mathbf{X}_Y \leftarrow \left\{X_i^- \,|\, Y_i^- = Y, i \in [n]\right\}$
6: $\mathbf{X}_{Y,\neg D} \leftarrow \left\{X_i^- \,|\, Y_i^- = Y, D_i^- \neq D, i \in [n]\right\}$
7: $\mathcal{L}_{\mathrm{aug}} \leftarrow -\log \frac{\exp s_\psi(A, Z)}{\sum_{A' \in \mathbf{A}} \exp s_\psi(A', Z)}$        $\triangleright -\mathrm{I}[A; Z]$
8: $\mathcal{L}_{\mathrm{supp}} \leftarrow -\log \frac{\sum_{X' \in \mathbf{X}_{Y,\neg D}} \exp e_\varphi(X')^T Z}{\sum_{X'' \in \mathbf{X}_Y} \exp e_\varphi(X'')^T Z}$    $\triangleright \mathrm{I}[Z; D\,|\,Y]$
9: **return** $\mathcal{L}_{\mathrm{C}^2\mathrm{AD}} = \mathcal{L}_{\mathrm{aug}} + \lambda \mathcal{L}_{\mathrm{supp}}$

---

## D    EXTENDED RELATED WORK

**Provably optimal representations.** Many previous work have theoretically studied advantages of representations in various two-stage settings (representation learning followed by standard training of predictors) by bounding downstream performance (e.g., Ben-David et al., 2007; Shamir et al., 2010; Saunshi et al., 2019). As learning theoretical bounds can be loose, it is hard to know whether they give the right insights into the problem. Our work instead proves the properties of *optimal* representations, which ensure best downstream performance. Those properties need to be approximated but give the right insights into what to aim for. This perspective and our proofs were inspired by Dubois et al. (2020) who gives sufficient conditions for optimal representations in supervised learning.

## E    EXPERIMENTAL DETAILS

### E.1    SCIENTIFIC

In both the scientific setting and the following bridge setting, we consider rather unrealistic setups for verifying our theory where we have access to labels from all domains. We can choose to directly minimize the risk $\mathrm{R}\left[Y \mid Z\right]$ with the cross-entropy loss (denoted as CE henceafter), or minimize $\mathrm{H}[A \mid Z]$ (i.e., maximize $\mathrm{I}[A; Z]$) with supervised augmentations as in Fig. 2b detailed below.

**Implementation of supervised augmentations**    When using supervised augmentations, for each sample we obtain its augmentations from within its label class across all domains. A constrastive loss with such augmentations will essentially reduce to the supervised contrast loss (SupCon, Khosla et al., 2020). In particular, for a single sample in a batch, all samples in the batch with the same labels can be used as the positives (could come from the same domain or different domains) and others as the negatives. In Khosla et al. (2020), two variants of SupCon loss were introduced for solving the issue of multi-positives depending on whether the summation over multi-positives was located inside (SupCon-In, Eq. (3) in Khosla et al. (2020)) or outside (SupCon-Out, Eq. (2) in Khosla et al. (2020)) the log. Though Khosla et al. (2020) chose SupCon-Out because it worked better than SupCon-In, we hypothesized that this is because SupCon-Out has an implicit bottleneck effect. Intuitively, SupCon-Out upper bounds SupCon-In and achieves its optima only if the logits with positive samples are all the same by Jensen's inequality, which may encourage positive samples from different domains to get clustered. Since this might confound with the effect of our bottlenecks, we chose to use SupCon-In though it performed slightly worse in out initial experiments. For the implementation of SupCon, we followed Khosla et al. (2020) except that no projection was used. Specifically, the temperature was set to 0.1, and normalization was applied when computing the logits.

In the scientific setting, we tried to simulate our theory to the greatest extent. In particular, we had two special considerations as detailed below:

**Eliminating empirical generalization**    As our theory focuses on the idealized domain generalization that assumes access to population distribution, we considered the setup where the empirical generalization was eliminated. Specifically, we treated the dataset as the population distribution and used the same dataset for training the encoder and training/evaluating the predictor. The ResNet-18 encoder was trained to 300 epochs without any regularization, using the Adam optimizer (Kingma & Ba, 2014) with a learning rate of 5e-5, a batch size of 192 (48 for each domain), and a cosine learning rate decay schedule.

**Worst-case approximation**    To approximate the worst-case source predictor, we included the target data with randomly assigned *wrong* labels to the training set for training the source predictor. The target data samples were down-weighted with a sample weight that maximizes the target risk while keeping the source risk close to optima (which is 0). We selected the sample weight by sweeping over $[10^{-10}, 1]$ with a logarithmic scale using CE-Base and SupCon-Base, as shown in Fig. 4. As the sample weight increases, the target log likelihood (neg. risk) first decreases and then increases. We hypothesized that the increasing trend was due to that the source performance was already not optimal (though not visible from the figure), thus we selected the weight close to the turning point and $10^{-5}$ seemed to be reasonable for both CE-Base and SupCon-Base. Although we did not adaptively select the sample weight for each setup due to the computational cost, the pre-specified sample turned

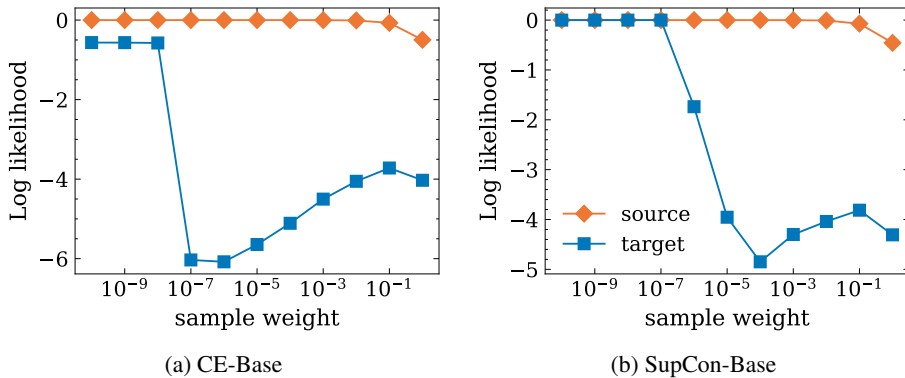

Figure 4: Sweeping the sample weight using CE-Base and SupCon-Base. We selected $10^{-5}$ which seemed to be reasonble for both cases.

out to be reasonable for all other losses and different $\lambda$ combinations. Furthermore, we also removed regularization when training the linear classifier and initialized the linear weight i.i.d. from $\mathcal{N}(0, 1)$.

Next, we provide other experimental details for reproducibility:

**Implementation of standard augmentations** We followed SimCLR (Chen et al., 2020) for implementing standard image augmentations. For a fair comparison between the cases when using standard augmentations (SimCLR) and supervised augmentations (SupCon), we kept the total batch size the same and also used the same configurations for computing the SupCon loss, i.e., temperature set to 0.1, no projection, and normalization applied.

**Details of Fig. 3c** In Fig. 3c, we considered different choices of augmentations. The 'Standard' augmentation implementation is described above (Appx. E.1). The 'Supervised' augmentation was essentially implemented using the SupCon loss as described in Appx. E.1. For other augmentations considered, we implemented them by *dropout* inter-domain supervised augmentations in SupCon. Specifically, for each sample in the batch, we randomly masked the samples from different domains (i.e., both inter-domain positives and negatives) i.i.d. with the specified dropout probability, while samples within the same domain were always kept. 'IntraDom' and 'ApproxDA' correspond to dropout probability 1 and 0.9, respectively. 'SingleDom' were implemented by dropout all inter-domain samples with probability 1 except for a fixed domain (the 'A' domain of PACS in our case).

### E.2 BRIDGE

In the bridge setting (see Appx. F.2), we aimed to bridge the gap between our theoretical setup to the practical setup. The main differences from the scientific setups are that the empirical generalization gap is considered and the average-case source predictor is used, as detailed below:

**Incorporating empirical generalization** In practice, empirical-generalization gap should also be considered besides the source-target generalization gap. Thus, we randomly split the PACS dataset to 80% training and 20% validation splits for each domain. The training splits were used to train both the encoder and the source predictor, and the validation splits were used for encoder and source predictor selection as well as evaluation on target domains. We used the ResNet-50 model as the encoder and initialized it from ImageNet pretrained model. The encoder was trained to a maximum of 50 epochs with a 1e-5 weight decay, using the Adam optimizer (Kingma & Ba, 2014) with a learning rate of 5e-5, a batch size of 112 (28 for each domain), and a cosine learning rate decay schedule.

**Using average-case source predictor** Instead of approximating the worst-case source predictor in the scientific setting, we considered the average-case[3] source predictor which is closer to the common practice. Specifically, we froze the encoder and trained a SVM classifier with L2 regularization on

---

[3]Here we have a slight abuse use of the phrase 'average-case' to distinguish from the 'worst-case' that we use in the scientific setting. In fact, the source predictor could be close to the 'best-case' since the max-margin classifier (SVM) was used.

the source training split. The regularization parameter was tuned over {1e-4, 1e-3, 1e-2, 1e-1, 1, 1e1, 1e2, 1e3} with the source validation accuracy.

Next, we provide other experimental details for reproducibility:

**Selection of** $\lambda$ For all different setups considered in bridge settings, the CAD bottleneck was used and the $\lambda$ was tuned over {1e-3, 1e-2, 1e-1, 1, 1e1} independently for each.

### E.3 DOMAINBED

**Datasets** We used non-MNIST datasets on DomainBed that were non-synthetic, including VLCS (Fang et al., 2013), PACS (Li et al., 2017), OfficeHome (Venkateswara et al., 2017), TerraIncognita (Beery et al., 2018), and DomainNet (Peng et al., 2019). For each dataset, we split it to 80%/20% training/validation set according to DomainBed.

**SSL-based models & Training** For all models based on pretrained SSL models (either CLIP-based or DINO-based) with finetuning in this experiment, we freezed the pretrained SSL model and added on top a 1-layer MLP with hidden size 1024, and residual connection. We used CLIP ResNet-50 (CLIP S) to obtain the best possible fair comparison with baselines from DomainBed, and CLIP ViT-B/32 (CLIP L) to achieve the best results. Note that the ResNet-50 model of CLIP S was modified as described in Radford et al. (2021) and contained 38M parameters (more than 23M of the original CLIP). The model was trained to 300 epochs for DomainNet and 50 epochs on other datasets (an epoch is defined as a single pass over the smallest domain according to DomainBed). No data augmentation was used and the temperature for scaling the logits in CAD was fixed to 0.05. We used the Adam optimizer with a 1e-5 weight dacay, and a cosine learning rate decay schedule. The hyperparameter search space is:

- Learning rate: discrete set {1e-4, 3e-4, 1e-3, 3e-3}
- Batch size: discrete set {128, 256, 512} for DomainNet and OfficeHome, and {64, 128, 256} for other datasets
- MLP dropout: discrete set {0., 0.1, 0.5}
- Learning rate warmup: discrete set {True, False}

**End-to-end models & Training** In Table 1, we also included an end-to-end trained model without any pretrained SSL models. We used exactly the same model architecture (the original ResNet-50, initialized from ImageNet pretrained model), training procedure and evaluation protocal as baselines on DomainBed. Importantly, the linear classifier was jointly trained with the encoder, and no refitting was applied. The model was trained to a maximum of 5000 steps on each dataset, and data augmentations were applied. The Adam optimizer was used without any particular learning rate schedule. The hyperparameter search space is (same as DomainBed except we added the temperature):

- Learning rate: log-uniform over [1e-5, 1e-3.5]
- Batch size: log-uniform over [8, 64] for DomainNet, and [8, $2^{5.5}$] for other datasets
- MLP dropout: discrete set {0., 0.1, 0.5}
- Weight decay: log-uniform over [1e-6, 1e-2]
- Temperature: discrete set {0.05, 0.1}

**Linear Probe Evaluation** In all the experiments except for the end-to-end training setup, we always followed the procedure of two-stage training, where we first trained the encoder with specified objectives, and then *refit* the classifier. For datasets except DomainNet, we fitted the SVM classifier and tuned the regularization parameter over {1e-4, 1e-3, 1e-2, 1e-1, 1, 1e1, 1e2, 1e3} with source validation selection. Since DomainNet was too large and SVM cannot fit it efficiently, we used the logistic regression classifier which was trained with a batch size 512, the Adam optimizer with a learning rate 5e-4 and early stopping. Note that an alternative was to just use the linear head fitted when training the representor (as we used CE loss with source labels), and we found this could work better than refitting since the classifier was less overfitted to the source domain. However, we didn't do that since we wanted to stick to the representation learning protocol with two-stage training. We

did that in our end-to-end training setup since we wanted it to be compeletely comparable to baselines on DomainBed (which did not do refitting).

**Selection of $\lambda$**    In our experiments, we treated $\lambda$ as a special hyperparamter. For each model, we used the same $\lambda$ selected on PACS on all datasets except DomainNet, because our bottleneck is fairly robust to the choice of $\lambda$. For the large-scale DomainNet dataset, we selected its $\lambda$ individually. The $\lambda$ values chosen for each model were:

- CLIP S: 1 on DomainNet and 1e-2 on other datasets
- CLIP L: 1e-1 on DomainNet and 1e-2 on other datasets
- DINO: 1e-1 on all datasets
- End-to-end ResNet-50: 1e-5 on all datasets

### E.4   LAION

**Model**    We used the CLIP L model (i.e., CLIP ViT-B/32) with an additional network on top for finetuning. The additional network were two blocks of 2-layer MLP, each with hidden size 2048, pre-activation batch normalization, residual connection, and dropout probability 0.1. Note that the original CLIP L model was frozen and only the additional network was trained.

**Dataset**    We used the LAION-400M dataset which contained 400 million image-text pairs for training. Though the dataset might not be as clean as the original CLIP training data (as evidenced by our experimental results), it was the largest publicly available image-text-pair dataset that we could get access to. As we froze the CLIP L model and only did finetuning, we used the 1TB preprocessed embeddings provided by LAION-400M[4]. No further preprocessing was applied.

**Training**    We used the image-text contrastive loss as introduced in Radford et al. (2021) for training model. The temperature was learnable which was initialized as 0.07 and clipped with a minimum 0.01. The model was trained for 1 epoch using the Adam optimizer with a batch size of 16384 and a cosine learning rate decay schedule. The learning rate was tuned over the set {3e-5, 1e-4, 3e-4, 1e-3, 3e-3, 1e-2} and the $\lambda$ value for the Ent bottleneck was tuned over {1e-3, 1e-2, 1e-1, 1, 1e1}.

**Evaluation**    For the evaluation on the ImageNet-related datasets, we followed a similar procedure in Radford et al. (2021), where a linear classifier was fitted on ImageNet using the model representations and evaluated on 7 natural distribution shift datasets. In particular, we fitted a logistic regression classifier with 1e-5 L2 regularization on ImageNet training set which was trained with a batch size 512, the Adam optimizer with a learning rate 3e-4 and early stopping. Note that this was different from Radford et al. (2021), where a logistic regression classifier was fitted using full-batch data with decent hyperparameter tuning, due to our computational budget. For evaluation on natural distribution shift datasets, we followed Taori et al. (2020) and used their released testbed[5]. The evaluation datasets and their abbreviations used in Table 2 were: ImageNetV2 (IN-V2, Recht et al., 2019), ImageNet-Sketch (IN-S, Wang et al., 2019), Youtube-BB (YT-BB, Shankar et al., 2019), ImageNet-Vid (IN-Vid, Shankar et al., 2019), ObjectNet (Barbu et al., 2019), ImageNet Adversarial (IN-A, Hendrycks et al., 2021), and ImageNet Rendition (IN-R Hendrycks et al., 2020).

---

[4]See `https://laion.ai/laion-400-open-dataset/` for details.
[5]`https://github.com/modestyachts/imagenet-testbed`

# F  ADDITIONAL EXPERIMENTAL RESULTS

## F.1  SCIENTIFIC

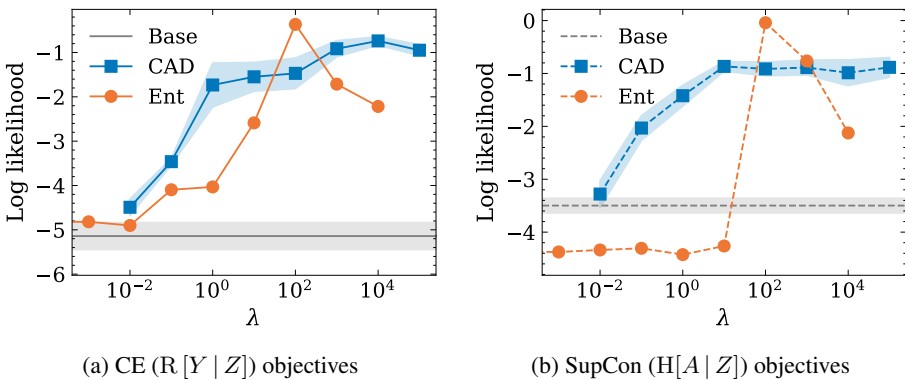

(a) CE (R $[Y \mid Z]$) objectives         (b) SupCon (H$[A \mid Z]$) objectives

Figure 5: The worst-case DG performance of Ent bottleneck is more sensitive to $\lambda$ than CAD

**What's the effect of $\lambda$ for different objectives on the worst-case DG performance?**  In Fig. 5, the worst-case target log likelihood versus $\lambda$ values for different objectives is shown. We found that Ent is much more sensitive to the choice of $\lambda$ than CAD, which was part of the reason why we used the latter in most of our experiments. Note that for SupCon-Ent with small $\lambda$ values, it was worse than SupCon-Base because of the discretization introduced by the Ent bottleneck, which we verified by observing that setting $\lambda = 0$ lead to similar results.

## F.2  BRIDGE

The scientific setup is closer to our theory than what we do in practice in that worst-case predictor was considered and empirical generalization gap was ignored. Here we bridged these gaps with a more practical setup. In particular, we split the PACS dataset to training and validation splits for each domain and considered the setting: the representor trains the encoder on all-domain training splits with a validation loss selection; the learner trains the SVM predictor (average-case) on the source training split which is selected over the source validation split, and evaluates on the validation splits of other target domains. The target validation accuracy averaged over all (source, target) setups was reported. For simplicity, we will use CE to denote the objective with the cross-entropy loss that uses labels to minimize R $[Y \mid Z]$, and SupCon for the contrastive loss that uses supervised augmentations to minimize H$[A \mid Z]$. We will use CAD in following experiments unless otherwise specified (chosen with initial experiments). Details in Appx. E.2.

Table 3: We repeated most empirical analysis (in the scientific setting) in the more practical bridge setting and observed similar results.

| Setup | Avg. target acc. |
|---|---|
| CE-Base | $95.9 \pm 0.5$ |
| CE-CAD | $96.7 \pm 0.2$ |
| CE-CAD (partial domains) | $82.6 \pm 0.5$ |
| SupCon-CAD | $96.7 \pm 0.4$ |
| SupCon-CAD (SingleDom) | $96.7 \pm 0.3$ |
| SupCon-CAD (ApproxDA) | $96.6 \pm 0.3$ |
| SupCon-CAD (IntraDom) | $96.2 \pm 0.7$ |
| SimCLR-CAD | $61.7 \pm 0.8$ |

**Does domain bottleneck improve the average-case DG performance?** Though our theory focuses on the worst-case DG, we empirically showed that adding bottlenecks to enforce support match can also improve the average-case DG performance by comparing CE-Base and CE-CAD in Table 3.

**What if the representor only has access to source domains?** Similar to what we did in the scientific setting, we considered the setup where one single domain is specified as the target domain and excluded from the training set of the representor and used for evaluation with source predictors trained on other domains. This is denoted as CE+CAD (partial domains) in Table 3, which is much worse then CE-CAD. This shows the necessity of getting access to target domain information for DG.

**What if the representor only has access to domain-agnostic augmentations?** In Table 3, we also compared SupCon-CAD which used supervised augmentations through the labels with CE-CAD and they achieved the same performance. This shows that the representor can still learn good representations without labels but only domain-agnostic augmentations in practice.

**Can we use standard augmentations?** In Fig. 2, we point out that standard augmentations are not domain-agnostic and thus not suitable for SSL with our objectives. We empirically showed this by using augmentations of SimCLR (see Appx. E.1 for details) with our objectives (SimCLR-CAD). In Table 3, we indeed observed that using standard augmentations performed much worse than using desired augmentations (SupCon-CAD).

**How do augmentations matter?** Besides investigating the 'Supervised' augmentations (SupCon-CAD) and 'Standard' augmentations (SimCLR-CAD) above, we also compared other three augmentations as in the scientific section. Specifically, we considered the 'SingleDom', 'IntraDom', and 'ApproxDA' augmentations. As shown in Table 3, SupCon-CAD (SingleDom) and (ApproxDA) maintained the DG performance but SupCon-CAD (IntraDom) was slightly worse (0.5 accuracy drop). We assumed the small gap was due to the specific dataset that we used (PACS). We did the same analysis on VLCS, and SupCon-CAD with 'Supervised', 'SingleDom', and 'IntraDom' augmentations gave $84.7 \pm 0.4$, $83.2 \pm 0.3$, and $77.5 \pm 2.3$, respectively. This shows the importance of using domain-agnostic augmentations in practice.

**Do standard augmentations affect source performance?** Previously, we showed that using standard augmentations hurt the DG performance measured by the average target accuracy. It is natural to ask whether using standard augmentations also hurt the source performance since we should also be interested in the 'effective robustness' (Taori et al., 2020). Thus we also reported the average source accuracy of SupCon-CAD and SimCLR-CAD which were $96.9 \pm 0.2$ and $90.1 \pm 0.2$, respectively. The source performance using standard augmentations was indeed worse, but if we consider the source-target gap which was 0.2 for SupCon-CAD and 28.4 for SimCLR-CAD, which still verified that the non-domain-agnostic standard augmentations were harder to force support match. To be even more convincing, we did the same analysis on VLCS, and the average source accuracy of SupCon-CAD and SimCLR-CAD were $86.6 \pm 0.1$ and $84.6 \pm 0.5$ which were fairly close, but the average target accuracy were $84.7 \pm 0.4$ and $57.5 \pm 1.7$, respectively.

## F.3 DOMAINBED

**Full result of Table 1** We included the full result of Table 1 with all baselines on DomainBed as in Table 4. We considered most representative baselines from DomainBed, most of which considered learning invariant representations or optimal classifiers across domains. Specifically, we included IRM (Arjovsky et al., 2019), GroupDRO (Sagawa et al., 2019), Mixup (Yan et al., 2020), CORAL (Sun & Saenko, 2016), MMD (Li et al., 2018a), DANN (Ganin et al., 2016), CDANN (Li et al., 2018b), and VREx (Krueger et al., 2021). We also included the result pretrained CLIP S model with a zero-shot classifier using text representations (CLIP S Zero Shot), which demonstrated better DG performance than CLIP S with linear probe. But we observed that it was outperformed by our CLIP S + CAD.

**What is the impact of CLIP pretraining?** To ensure that our gains are *not only* due to a novel CAD bottleneck, but the synergy between enforcing support constraint and using desired SSL models, we investigated CAD using the standard DomainBed protocol denoted as CAD in the table. It shows that CAD on its own performs similarly with DomainBed baselines (see Table 4 for a full comparison).

Table 4: Full results on DomainBed with 'oracle selection' method.

| Algorithm | VLCS | PACS | OfficeHome | TerraIncognita | DomainNet |
|---|---|---|---|---|---|
| ERM | $77.6 \pm 0.3$ | $86.7 \pm 0.3$ | $66.4 \pm 0.5$ | $53.0 \pm 0.3$ | $41.3 \pm 0.1$ |
| IRM | $76.9 \pm 0.6$ | $84.5 \pm 1.1$ | $63.0 \pm 2.7$ | $50.5 \pm 0.7$ | $28.0 \pm 5.1$ |
| GroupDRO | $77.4 \pm 0.5$ | $87.1 \pm 0.1$ | $66.2 \pm 0.6$ | $52.4 \pm 0.1$ | $33.4 \pm 0.3$ |
| Mixup | $78.1 \pm 0.3$ | $86.8 \pm 0.3$ | $68.0 \pm 0.2$ | $\mathbf{54.4 \pm 0.3}$ | $39.6 \pm 0.1$ |
| CORAL | $77.7 \pm 0.2$ | $87.1 \pm 0.5$ | $68.4 \pm 0.2$ | $52.8 \pm 0.2$ | $41.8 \pm 0.1$ |
| MMD | $77.9 \pm 0.1$ | $87.2 \pm 0.1$ | $66.2 \pm 0.3$ | $52.0 \pm 0.4$ | $23.5 \pm 9.4$ |
| DANN | $79.7 \pm 0.5$ | $85.2 \pm 0.2$ | $65.3 \pm 0.8$ | $50.6 \pm 0.4$ | $38.3 \pm 0.1$ |
| CDANN | $79.9 \pm 0.2$ | $85.8 \pm 0.8$ | $65.3 \pm 0.5$ | $50.8 \pm 0.6$ | $38.5 \pm 0.2$ |
| VREx | $78.1 \pm 0.2$ | $87.2 \pm 0.6$ | $65.7 \pm 0.3$ | $51.4 \pm 0.5$ | $30.1 \pm 3.7$ |
| CAD | $78.0 \pm 0.1$ | $87.3 \pm 0.2$ | $67.0 \pm 0.5$ | $53.5 \pm 0.9$ | $41.5 \pm 0.1$ |
| DINO + CAD | $69.6 \pm 0.6$ | $76.1 \pm 0.1$ | $56.9 \pm 0.5$ | $25.9 \pm 1.2$ | $33.6 \pm 0.1$ |
| CLIP S | $81.1 \pm 0.5$ | $90.3 \pm 0.2$ | $70.6 \pm 0.1$ | $29.6 \pm 0.8$ | $47.7 \pm 0.0$ |
| CLIP S (Zero-Shot) | $80.9 \pm 0.1$ | $91.8 \pm 0.1$ | $70.4 \pm 0.2$ | $19.1 \pm 0.1$ | $46.9 \pm 0.0$ |
| CLIP S + Base | $81.3 \pm 0.5$ | $91.2 \pm 0.3$ | $70.6 \pm 0.1$ | $36.4 \pm 0.7$ | $46.8 \pm 0.2$ |
| CLIP S + CAD | $\mathbf{82.3 \pm 0.3}$ | $92.0 \pm 0.2$ | $71.9 \pm 0.2$ | $36.2 \pm 0.8$ | $48.8 \pm 0.1$ |
| CLIP L | $80.7 \pm 0.4$ | $93.7 \pm 0.8$ | $79.6 \pm 0.1$ | $36.9 \pm 0.6$ | $52.8 \pm 0.1$ |
| CLIP L + CAD | $81.6 \pm 0.1$ | $\mathbf{94.9 \pm 0.3}$ | $\mathbf{80.0 \pm 0.2}$ | $40.6 \pm 1.1$ | $\mathbf{53.7 \pm 0.1}$ |

**Why 'oracle' selection?**    In the main body, we provided the results with 'oracle selection' which was the closest to our theory among the model selection methods in DomainBed (in the sense that we needed target domain information to achieve IDG). Here, we also provided results with 'source validation' selection in Table 5. Source validation selection relies on the assumption that source and target data follow similar distributions (Gulrajani & Lopez-Paz, 2021) thus source and target accuracy are highly correlated, which is not really true in practice. We found some issues with source validation selection results:

- The selected model with the highest source validation accuracy tends to overfit the source domain, thus possibly leads to worse performance on the target domain. This can be probed by the fact that the finetuned CLIP models (CLIP + Base or CLIP + CAD) were generally worse than the original CLIP model;

- Selecting model with source validation accuracy tends to diminish the effect of bottlenecks. This can be seen by the fact that the gap between CLIP + Base and CLIP + CAD of source validation selection is much smaller than that of oracle selection;

- The source accuracy is not a good indicator of target accuracy thus its result has a larger variance.

## F.4    LAION

**Evaluation results on DomainBed**    We included the evaluation results of trained models on DomainBed in Table 6, where we followed exactly the same linear evaluation protocal discussed in Appx. E.3. We observed similar results as Table 2: the CLIP L model trained with the Ent bottleneck on LAION (Tuned w/ Ent) outperformed the one without (Tuned w/o Ent) on all DomainBed datasets, but slightly underperformed the original CLIP L model (which might be due to quality of the LAION-400M dataset).

Table 5: Results on DomainBed with 'source validation' selection. Source validation selected model tends to overfit more to the source domain and diminish the effect of bottlenecks.

| Algorithm | VLCS | PACS | OfficeHome | TerraIncognita | DomainNet |
|---|---|---|---|---|---|
| ERM | $77.5 \pm 0.4$ | $85.5 \pm 0.2$ | $66.5 \pm 0.3$ | $46.1 \pm 1.8$ | $40.9 \pm 0.1$ |
| IRM | $78.5 \pm 0.5$ | $83.5 \pm 0.8$ | $64.3 \pm 2.2$ | $47.6 \pm 0.8$ | $33.9 \pm 2.8$ |
| GroupDRO | $76.7 \pm 0.6$ | $84.4 \pm 0.8$ | $66.0 \pm 0.7$ | $43.2 \pm 1.1$ | $33.3 \pm 0.2$ |
| Mixup | $77.4 \pm 0.6$ | $84.6 \pm 0.6$ | $68.1 \pm 0.3$ | $47.9 \pm 0.8$ | $39.2 \pm 0.1$ |
| CORAL | $78.8 \pm 0.6$ | $86.2 \pm 0.3$ | $68.7 \pm 0.3$ | $47.6 \pm 1.0$ | $41.5 \pm 0.1$ |
| MMD | $77.5 \pm 0.9$ | $84.6 \pm 0.5$ | $66.3 \pm 0.1$ | $42.2 \pm 1.6$ | $23.4 \pm 9.5$ |
| DANN | $78.6 \pm 0.4$ | $83.6 \pm 0.4$ | $65.9 \pm 0.6$ | $46.7 \pm 0.5$ | $38.3 \pm 0.1$ |
| CDANN | $77.5 \pm 0.1$ | $82.6 \pm 0.9$ | $65.8 \pm 1.3$ | $45.8 \pm 1.6$ | $38.3 \pm 0.3$ |
| VREx | $78.3 \pm 0.2$ | $84.9 \pm 0.6$ | $66.4 \pm 0.6$ | $46.4 \pm 0.6$ | $33.6 \pm 2.9$ |
| CAD | $78.0 \pm 0.5$ | $85.2 \pm 0.9$ | $67.4 \pm 0.2$ | $47.3 \pm 2.2$ | $41.0 \pm 0.1$ |
| DINO + CAD | $68.9 \pm 0.9$ | $75.4 \pm 0.5$ | $56.4 \pm 0.7$ | $23.6 \pm 1.2$ | $31.0 \pm 2.3$ |
| CLIP S | $81.1 \pm 0.5$ | $90.3 \pm 0.2$ | $70.6 \pm 0.1$ | $29.6 \pm 0.8$ | $47.7 \pm 0.0$ |
| CLIP S (Zero-Shot) | $80.9 \pm 0.1$ | $91.8 \pm 0.1$ | $70.4 \pm 0.2$ | $19.1 \pm 0.1$ | $46.9 \pm 0.0$ |
| CLIP S + Base | $81.4 \pm 0.4$ | $89.6 \pm 0.7$ | $70.4 \pm 0.2$ | $30.9 \pm 2.2$ | $44.6 \pm 1.6$ |
| CLIP S + CAD | $81.2 \pm 0.6$ | $90.0 \pm 0.6$ | $70.5 \pm 0.3$ | $30.3 \pm 0.9$ | $45.5 \pm 2.1$ |
| CLIP L | $80.6 \pm 0.7$ | $93.5 \pm 0.8$ | $79.4 \pm 0.2$ | $37.5 \pm 0.7$ | $50.1 \pm 1.1$ |
| CLIP L + CAD | $80.8 \pm 0.7$ | $93.5 \pm 0.7$ | $79.7 \pm 0.2$ | $37.4 \pm 1.2$ | $51.7 \pm 1.4$ |

Table 6: Finetuning CLIP L on LAION with an entropy bottleneck performs better on DomainBed than finetuning without.

| Algorithm | VLCS | PACS | OfficeHome | TerraIncognita | DomainNet |
|---|---|---|---|---|---|
| CLIP L | $80.7 \pm 0.4$ | $93.7 \pm 0.8$ | $79.9 \pm 0.1$ | $36.9 \pm 0.6$ | $52.8 \pm 0.1$ |
| Tuned w/o Ent | $79.2 \pm 0.7$ | $93.4 \pm 0.3$ | $77.2 \pm 0.5$ | $36.1 \pm 0.4$ | $51.2 \pm 0.1$ |
| Tuned w/ Ent | $80.7 \pm 0.4$ | $94.3 \pm 0.8$ | $78.2 \pm 0.2$ | $36.8 \pm 0.4$ | $52.2 \pm 0.1$ |

