# OpenReview forum: "Optimal Representations for Covariate Shift"
_ICLR.cc/2022/Conference — ICLR 2022 Poster_

### Official Review · Reviewer_1s9j · 2021-11-01

**Correctness:** 4
**Technical Novelty And Significance:** 3
**Empirical Novelty And Significance:** 3
**Recommendation:** 8
**Confidence:** 3

**Main Review:**

*Strong point(s) of the paper*

- The paper contributes some lucid new theoretical insights to the problem of encoder design that is conducive to trained models which are domain robust, and does an effective job of linking this basic theory up with practical training procedures.

- The writing is solid. High-level theoretical points are quite clear in the main paper (details in the appendix), the relation to existing work is described, and the experimental results are distilled into new questions/insights which are easily parsed by the reader.

*Weak point(s) of the paper*

- The theoretical results are stimulating, but they are quite idealized and leave a substantial gap between what we can know and check in practice, and what we want to know formally. For example, while the notion of a domain-covering augmentation is very simple mathematically, but one expects that evaluating whether we have something close to such an augmentation in practice becomes quite challenging when tasks are difficult to describe in words or in simple visual terms.

*Questions*

- Most of the notation is immaculate, but I would like to confirm about  $\\mathrm{R}_{h}^{d}[Y \\,|\\, Z]$. Is the assumption that $Z \\sim p\_{Z \\,|\\, X}$ and the expectation is now with respect to $Y$, $X$, and the randomness in sampling $Z$? (instead of just $Y$ and $X$)

**Summary Of The Paper:**

At a high level, this work considers the problem of designing machine learning systems that generalize well even when the "domain" (i.e., the data-generating process) under which the system is tested (the "target") does not match that under which it was trained (the "source"). More specifically, the authors look to provide an answer to the question of what makes a "good" representation or encoding (a stochastic transformation of the original features), from the perspective of achieving a small risk (expected loss) on the target domain, assuming the learner only has data (and thus representations) from the source domain.

To this question, they provide one answer by showing that in a generalized covariance shift scenario, a natural notion of representation optimality (their "IDG optimality") is only possible if a representation minimizes the (best) risk on the average target distribution, while ensuring that the support of the representations is constant across domains. This is obviously a strong requirement, and they show that one cannot expect to satisfy it without domain knowledge going beyond the source.

They also show that there is some hope for special cases of domains in which we have augmentations that preserve label information and "cover" the support of the input distribution across all domains. In such a case, the authors show that their strong optimality requirement can be satisfied by designing the encoder to maximize the mutual information with the augmented data. Relaxing this objective leads to objectives that are more practical, and the authors complement their theoretical analysis with a rather in-depth set of empirical tests that evaluate the efficacy of their methodology for representation learning.

**Summary Of The Review:**

The problem is fundamental to machine learning, the theory developed by the authors is stimulating and to the best of my knowledge new, the empirical analysis is informative, and the writing throughout is very strong. I vote to accept.

---

> ### Author Response · Authors · 2021-11-18
> **Minor**
>
> > *Clarification on notation*
>
> Yes, that’s correct. Specifically, the risk $R_h^d[Y|Z]$ is taken expectation over the joint distribution $p_{Y,Z|d}$ which is induced from $p_{Y,X|d}$ by encoder $p_{Z|X}$.

---

> ### Author Response · Authors · 2021-11-18
> **Domain-agnostic augmentations can be hard to find, but we're the first to show which augmentation properties have differential impacts on DG**
>
> > *“one expects that evaluating whether we have something close to such an augmentation in practice becomes quite challenging when tasks are difficult to describe in words or in simple visual terms”*
>
> We agree that evaluating whether an augmentation is appropriate may be challenging, and require domain-specific knowledge on the part of the researcher. Nevertheless, our work is the first to even distinguish between types of augmentations as it relates to domain generalization, and we believe that this brings value even if it may be difficult to evaluate in some cases. Our characterization of domain-agnostic (renaming of domain-covering) augmentations provides specific guidance on what augmentations should satisfy for them to be appropriate for use in learning robust representations. As a concrete example, in our paper, this directly indicates that text-image augmentations lead to more robust representations than standard image augmentations since they are much more domain-agnostic. We note that there could be more such augmentations that we can explore in practice, especially for multi-modal augmentations, e.g., [1]. We would like to refer the reviewer to [our response to Reviewer hrZK](https://openreview.net/forum?id=Rf58LPCwJj0&noteId=4FOTv9RJyF) for more information.
>
> [1] Wang, Luyu, et al. "Multimodal Self-Supervised Learning of General Audio Representations." arXiv preprint arXiv:2104.12807 (2021).

---

> ### Author Response · Authors · 2021-11-18
> **On the gap between theory and practice**
>
> > *”The theoretical results are stimulating, but they are quite idealized and leave a substantial gap between what we can know and check in practice”*
>
> We agree that there is still a gap between theory and practice. For example, we only study idealized domain generalization: perfect optimizers, perfect augmentations, optimal representations, and focus on domain generalization rather than sample generalization... We hope that our new tools and framework lay the foundations to allow us and others to extend our theoretical results in future work.

---

### Official Review · Reviewer_b5Jo · 2021-11-02

**Correctness:** 3
**Technical Novelty And Significance:** 3
**Empirical Novelty And Significance:** 2
**Recommendation:** 5
**Confidence:** 3

**Main Review:**

Strength

- Theoretical analysis on the conditions for idealized domain generalization

Weaknesses

- Writing is poor
- Empirical improvements come from external datasets rather than the method itself

The paper focuses on learning representations for domain generalization. The main idea is to learn a representation that matches the support across domains while maintaining discriminative information (Thm.1). Starting from idealized domain generalization risk (page 2), the paper goes through a long path of derivations and assumptions to derive practical objectives. The theoretical understanding is interesting, but the exposition is poor and empirical results may not be as promising as it seems.

1. Technicalities (and required clarifications for some parts)

- It is surprising to see that the current version does not discuss hypothesis class in question. For example, after (2), what's the candidate hypothesis set for h (argmin_h) when defining the set of source risk minimizers? The hypothesis class usually plays a central role in domain adaptation (e.g., Cortes et al. (2011; 2019)). Additionally, a motivating example would be great for introducing IDG risk and why we should minimize it.
- Page 3 mentioned different scenarios of Assumption 1 (GCS) for different losses like log-loss, 0-1 loss and MSE. Can you elaborate or provide some derivations?
- After Thm.1, it is claimed that "risk minimization using a single domain is as good as performing risk minimization on all domains". However, both R_{IDG} and R involve expectations over domains or pairs of domains. None is an optimization over a single domain.
- It is not clear how "the key requirement" can be satisfied by the given augmentation. Specifically, how to design such augmentation, other than CLIP?

Ref:

- Cortes, C. and Mohri, M. Domain adaptation in regression. In International Conference on Algorithmic Learning Theory, 2011.
- Cortes, C., Mohri, M., and Medina, A. M. Adaptation based on generalized discrepancy. The Journal of Machine Learning Research, 2019.

2. The writing is poor. Some parts of the paper are challenging to follow

- Notations are not consistent. Sometimes d_t (after Eq.(1)), sometimes D_t (Eq.(2), Theorem 1 etc)
- This paper fails to properly introduce the necessary background on CLIP.
- Exposition in Sec.4.1 is poor. It would be much better if the paper can provide a running example about A, X and Z.
    - A is said to be a RV. If I understand it correctly, an augmentation A can take value in, for example, {image rotation, cropping, mirroring} as shown in Fig.2a. Then p_{A|x} is the distribution of the above three augmentations conditioned on x. It is not clear why p_{A|x}=p_{A|x'} implies they have the same Bayes predictions. If two images are destined to be mirrored with probability one, then they must have the same prediction? If A is supposed to be the actual mirror image of x, then it is more than an arbitrary RV and should be specified more precisely. It is unclear where this "key requirement" comes from.
    - What is the task/classes in Fig.2? Is it classifying "floppy ear" versus "pointed-eared"? If so, the augmentations in Fig.2b look arbitrary and do not make sense.
    - "Enforcing the support constraint for augmented data (X, A)", but the constraint of Eq.(5) does not involve the augmentation A.
- Sec.4.2
    - What is the definition of the domain bottleneck B[Z, D]? And why it is related to the support constraint in Eq.(5)?
    - What is \mathcal{A} after Eq.(7)?

3. Experiments

- What is the "fixed domain" for SingleDom? And why it is DC?
- For approximate DC, what's the result if SingleDom is used instead of Supervised?
- The improvement in Table 1 seems to be mainly due to the fact that CLIP is pre-trained using a much larger dataset. Thus the proposed methods are only marginally better than vanilla CLIP if I understand it correctly. The results on TerraIncognita can justify this. To add some discussion, how can we ensure domain-covering in practice when even CLIP may not have sufficient coverage?

**Summary Of The Paper:**

This work focuses on learning representations for domain generalization. It proposes to minimize the idealized domain generalization risk (IDG risk; defined on page 2). Under several assumptions (importantly, domain covering augmentations), the IDG risk objective can be altered (Thm.1 -> Prop.2 -> Eq.(6)) to a more practical one. With additional variational techniques (Sec.4.2), the proposed algorithms approximately minimize conditional entropy and domain bottleneck in (6). Experiments on standard benchmarks show that the proposed method can out-perform SOTA alternatives.

**Summary Of The Review:**

In summary, the paper theoretically analyzes the conditions under which domain generalization would be possible. However, the writing is not clear at places and the empirical results are not very convincing.

---

> ### Author Response · Authors · 2021-11-18
> **Clarification for other technical confusions and concerns**
>
> > *”It is surprising to see that the current version does not discuss hypothesis class in question”*
>
> We agree that hypothesis class is very important. We consider the universal hypothesis class for simplicity in our paper. However, we can potentially generalize our theoretical results to a constrained hypothesis class. Briefly, the sufficiency and existence results would still hold straightforwardly; while for the necessity to hold, the support constraint can likely be weakened. We discuss it in our conclusion and leave it for future work.
>
>
> > *”Page 3 mentioned different scenarios of Assumption 1 (GCS) for different losses like log-loss, 0-1 loss and MSE. Can you elaborate or provide some derivations?”*
>
> Our GCS assumption concerns the same Bayes predictor $f^*$ across domains, and leads to specific forms for different losses. Suppose the label distribution for a specific sample $x$ is $p_{Y|x}$. For log-loss, the Bayes prediction is $f^*(x)=p_{Y|x}$. For 0-1 loss, $f^*(x)=\arg\max_{y} p_{Y|x}(y)$. For MSE loss, $f^*(x)=\mathbb{E}[p_{Y|x}]$. For MSE, our GCS thus assumes that $\mathbb{E}[p_{Y|x,d}] = \mathbb{E}[p_{Y|x}]$. Standard covariate is the most stringent GCS and is recovered in the log loss case $p_{Y|x,d}=p_{Y|x}$.
>
>
> > *”Notations are not consistent. Sometimes d_t (after Eq.(1)), sometimes D_t (Eq.(2), Theorem 1 etc)”*
>
> We respectfully clarify that it is not notation inconsistency. We use uppercase letters as random variables and lowercase letters as their realizations, which is the common practice in probability theory.
>
>
> > *”This paper fails to properly introduce the necessary background on CLIP.”*
>
> We apologize for cutting down the background on CLIP due to space constraint, and have added more descriptions throughout the paper.
>
>
> > *”"Enforcing the support constraint for augmented data (X, A)", but the constraint of Eq.(5) does not involve the augmentation A”*
>
> We agree the statement is not very clear. We wanted to state that, using only the augmented data $(X, A)$, it is still not clear how to optimize Eq. (5) since the support constraint requires access to domain information $D$. We have rephrased this statement in the revised paper.
>
>
> > *”What is the definition of the domain bottleneck B[Z, D]? And why it is related to the support constraint in Eq.(5)?”*
>
> As described in Sec 4.2, the domain bottleneck $B[Z, D]$ is introduced for enforcing the support match constraint in Eq. (5), and minimizing $B[Z,D]$ while maximizing $I[A;Z]$ achieves the support constraint. We can thus derive a family of SSL objectives with different valid $B[Z, D]$; and we introduce two instantiations, i.e., the CAD bottleneck $B[Z,D]=I[Z;D]$ and the Ent bottleneck $B[Z,D]=H[Z]$.
>
>
> > *“What is \mathcal{A} after Eq.(7)?”*
>
> $\mathcal{A}$ is the sample space of augmentations.
>
>
> > *”What is the "fixed domain" for SingleDom? And why it is DC?”*
>
> The fixed domain is a pre-fixed domain (in our experiments the “A” domain of the PCAS dataset) for sampling the augmentations. Specifically, for all domains, the augmentation is uniformly sampled from those samples with the same label *only in this domain*. Thus the set of augmentation distributions is the same for all domains, which is the uniform distributions over samples with the same label for different labels in this domain.
>
> > *“For approximate DC, what's the result if SingleDom is used instead of Supervised?”*
>
> We have run this setup (ApproxDC with SingleDom, denoted as ApproxDC 2), as shown below with other setups. We see that it is close to ApproxDC with Supervised and SingleDom, but still much better than other non-DC ones.
>
> | Aug. type  | Log likelihood   |
> | ------------- |-------------|
> |  Supervised    | $-0.82 \pm 0.13$ |
> |  SingleDom    | $-0.97 \pm 0.12$ |
> |  ApproxDC    | $-0.91 \pm 0.15$ |
> |  ApproxDC 2    | $-1.14 \pm 0.22$ |
> |  IntraDom    | $-7.13 \pm 0.30$ |
> |  Standard    | $-7.50 \pm 0.23$ |

---

> ### Author Response · Authors · 2021-11-18
> **Designing augmentations depends on specific domains, but we're the first to show which augmentation properties have differential impacts on DG**
>
> > *“It is not clear how "the key requirement" can be satisfied by the given augmentation. Specifically, how to design such augmentation, other than CLIP?”*
>
> Our result provides guidance on what properties the augmentations should have to learn robust representations in practice. There are two requirements: domain-agnostic and invariance of the Bayes predictor. Invariance of the Bayes predictor essentially holds for all typical augmentations. Domain-agnostic is the more challenging requirement and we point out that text-image essentially satisfies it. We focused on text-image augmentations given their availability online and the recent CLIP results, but many other multimodal augmentations would likely be domain-agnostic, e.g., [2]. Still, it’s hard to give very generic answers to the question of designing such augmentations, because it will depend heavily on the domain. Nevertheless, our work is the first to even distinguish between types of augmentations as it relates to domain generalization. For more information see [our response to Reviewer hrZK](https://openreview.net/forum?id=Rf58LPCwJj0&noteId=4FOTv9RJyF).
>
>
> [2] Wang, Luyu, et al. "Multimodal Self-Supervised Learning of General Audio Representations." arXiv preprint arXiv:2104.12807 (2021).

---

> ### Author Response · Authors · 2021-11-18
> **Clarification about augmentations**
>
> We respectfully point out that the reviewer seems to be confused about our definition of augmentations. We apologize for the confusion and would like to clarify it here.
>
> > *”If I understand it correctly, an augmentation A can take value in, for example, {image rotation, cropping, mirroring} as shown in Fig.2a.”*
>
> This is not true. By our definition, the augmentation $A$ is not the random augmentation operator but the *actual random augmented example* after applying the augmentation operator to the input, $\mathcal{A}$ is the sample space of augmented examples. This is why $A$ is sampled conditionally on the input $X$. For example, for standard image augmentation, the augmentation space $\mathcal{A}=\mathcal{X}$ is the sample space of images, $A$ is the augmented image after applying random cropping, rotation, etc. For image-text augmentations, $\mathcal{A}$ is the sample space of text captions, $A$ is the text obtained by sampling descriptive image captions.
>
>
> > *“It is not clear why p_{A|x}=p_{A|x'} implies they have the same Bayes predictions”*
>
> This is the assumption we make on the augmentation. It holds for most cases and is in fact the (implicit) assumption of recent self-supervised learning methods, e.g., in [1]. In standard settings where $\mathcal{A}=\mathcal{X}$, this essentially means that if two images can be augmented to one another they must have the same label.
>
>
> > *“What is the task/classes in Fig.2? Is it classifying "floppy ear" versus "pointed-eared"?”*
>
> In Fig. 2, we are visualizing the representation learned by using different kinds of augmentations to highlight the effectiveness of domain-agnostic augmentations, using a single class (“dog”) as illustration. The downstream task is not shown in the figure, but it could be, e.g., classifying “dog” vs “cat”.
>
> We clarify that "floppy ear" versus "pointed-eared" is not the task label but the text augmentations of different dog images. The key point here is that the dog images in different domains with similar semantic meanings (sketch and photo) are mapped to the same text caption which helps “bucket” their representations together, as opposed to standard image augmentations.
>
>
> [1] Arora, Sanjeev, et al. "A theoretical analysis of contrastive unsupervised representation learning." arXiv preprint arXiv:1902.09229 (2019).

---

> ### Author Response · Authors · 2021-11-18
> **We clarified our two-stage setup for learning robust representations for IDG**
>
> > *”Additionally, a motivating example would be great for introducing IDG risk and why we should minimize it.”*
>
> As a concrete example, suppose the task is traffic sign prediction and we are given unlabeled data from different countries. We aim to learn robust representations for these data such that: given labeled data from any of these countries, the predictor we train on these data is guaranteed to perform well in the worst case in any other countries.
>
> We find the basic setup of the preceding example to be a natural setting for DG, and our IDG risk for representation learning is motivated by a two-stage abstraction of the preceding example. In the first stage, we learn an encoder $p_{Z|X}$ to map the input $X$ to representation $Z$. In the second stage, the encoder $p_{Z|X}$ is *fixed* and a random source domain $D_s$ is given for training the predictor $h$. We would like to ensure that all possible optimal source predictors achieve a low risk on any random target domain $D_t$, thus motivating us to evaluate the worst-case target risk over all source optimal predictors. Our final IDG risk is obtained by taking expectation over source-target pairs.
>
> > *”After Thm.1, it is claimed that "risk minimization using a single domain is as good as performing risk minimization on all domains". However, both R_{IDG} and R involve expectations over domains or pairs of domains. None is an optimization over a single domain”*
>
> $R_{IDG}[Y|Z^*]=R[Y|X]$ means that for each pair of source and target domain, the predictor $h$ **trained only on source** is as good as if it was trained on all domains. In other words, risk minimization using a single domain and $Z^*$ is as good as performing risk minimization on the target domain from inputs $X$. Although $R_{IDG}$ involves an expectation over all domains, it is just for *evaluating* the learned representation in the second stage. We have clarified it in the paper.

---

> ### Author Response · Authors · 2021-11-18
> **The gains of CLIP are mainly due to its augmentations, and the improvement of our bottlenecks is large even without end-to-end training**
>
> > *”The improvement in Table 1 seems to be mainly due to the fact that CLIP is pre-trained using a much larger dataset. Thus the proposed methods are only marginally better than vanilla CLIP if I understand it correctly. The results on TerraIncognita can justify this”*
>
> We respectfully disagree with the point that “the improvement is mainly due to the fact that CLIP is pre-trained using a much larger dataset”. We believe that differential performance of SSL methods is explained by the choice of augmentations. We demonstrate this point empirically. In particular, we show that using domain-agnostic (we renamed “domain-covering” to “domain-agnostic” to avoid confusion, see general response) augmentations leads to much better DG performance than non-domain-agnostic standard augmentations, in both our scientific setup (Fig. 3(c), -1.0 vs - 7.5 target log lik.) and our non-idealized bridge setup (Table 3, 96.7 vs 61.7 target acc.). These experiments clearly demonstrate that augmentations matter in settings where the dataset is the held constant.
>
> While we would have loved to run experiments to distinguish the impact of CLIP’s dataset, this is not possible because we don’t have access to it. Instead we compared CLIP + CAD (our method) to CLIP+base and raw CLIP to demonstrate that CLIP itself can still be improved. We believe the improvements that we found using our bottlenecks are more significant and promising than what the percentages show. We include our full results with all previous DG baselines in Table 4; none of them could really outperform naive ERM and even the best algorithm (CORAL) only achieves about 0.6% improvement on average. Even the “DomainBed SOTA” in Table 1 which is the **best** result selected for each dataset **separately**, only outperforms ERM by 0.5~2%. In contrast, our CLIP + CAD **consistently** outperforms CLIP + Base and raw CLIP by a similar margin. The improvement is obtained by finetuning the last layers and we believe that it could be even more significant if CLIP was trained end-to-end with our bottlenecks.

---

### Official Review · Reviewer_hrZK · 2021-11-05

**Correctness:** 4
**Technical Novelty And Significance:** 3
**Empirical Novelty And Significance:** 3
**Recommendation:** 6
**Confidence:** 3

**Main Review:**

Novelty of the contribution: It seems to me that the main contribution of the paper is to provide informative formulations that characterize the optimal representation, which provide guidance on designing a representation learning scheme under covariate shifts. This is a potentially important result, though I am not sure how much this distinguishes from the literature. From the related work discussed in Section 5, it has been known that it is sufficient to match the support across domains. The authors emphasize that the condition they proposed is also necessary, but how non-trivial is the necessity? Intuitively, if the representation of the source and target could have different support, then it is hard to guarantee good performance in the target. Perhaps the main contribution here is to make this intuition concrete, or perhaps crafting the minimax optimality in IDG? The authors might need to clarify and provide a more convincing argument to support their novelty.

Requiring access to lots of information: As discussed by the authors, one limitation of their framework is to assume access to an overly demanding amount of information that seems not practical. Would a more meaningful framework be to look at representation trained only on the source domain (as in the setting of Section 6.2 which is more practical) and some degree of information from other domains, and consider the characterization of optimality under this setting. In this latter setting, is there a way to generalize Theorem 1 and if so how much would the characterization change? (I understand a full answer may be difficult, but at least some indications to boost my confidence would be helpful; see also the next point that has a similar flavor)

Assumption on the domain-covering augmenter: Proposition 2 relies on the existence of a domain-covering augmenter. It is argued that image-text augmentations are nearly domain-covering, but it seems hard to expect that they are strictly domain-covering. So there appears a gap in the theory: The theory requires a strict domain-covering augmenter, while in practice this can only be nearly satisfied. Nevertheless, I think the proposition gives insights how the augmenter could be useful..

Numerical results: It seems that the benefit of domain bottlenecks is not very significant. In Table 1, sometimes the improvement of CLIP S/L + CAD over CLIP S/L is not significant enough, e.g., 94.7\pm 0.4 of CLIP L+CAD vs 93.7\pm 0.8 given in the PACS column. The only significant improvement is seen in the DomainNet column where the improvement is about 1%.

A few minor issues:
Page 1 the first summary point: it is said that the paper provides “all” objectives that have the correct optima. Why couldn’t there exist other objectives not given in this paper?
Page 3, the paragraph after Thm 1: it seems that the cross references to Thm 2 should be corrected to refer to Thm 1 (and similarly for other instances).
Page 3, some minor typos in the paragraph after Proposition 1: e.g., “to to”, “worst IDG”.
Table 1: it would be better to explain the level of the confidence interval (CI) in each entry, e.g., are they 95% CIs?

**Summary Of The Paper:**

The paper studies representation learning under covariate shift. Under the IDG setting and the proposed assumptions, the paper gives a so-called variational characterization of the optimal representation. This characterization shows that the optimal representation should remain discriminative while has the same support across domains. It is argued that without any target information, no representation can do uniformly well over constant representation, thus supporting the necessity of target knowledge. The paper provides practical objectives of the proposed variational characterization by self-supervised learning using domain-covering augmentations. The proposed representation learning scheme is tested on several datasets.

**Summary Of The Review:**

The paper provides insights on the optimal representation under covariate shifts, which is a potentially important problem. I think the authors did concrete work to develop the theory and set up the numerical experiments, though there are also limitations discussed in the main review.

---

> ### Author Response · Authors · 2021-11-18
> **Minor answers**
>
> > *“it is said that the paper provides “all” objectives that have the correct optima. Why couldn’t there exist other objectives not given in this paper?”*
>
> We apologize for the confusion and have modified the claim in the paper. We meant that our variational objective leads to the exact set of all optimal representations for IDG since it is both sufficient and necessary. So all other objectives should have optimas that satisfy our conditions; our framework thus gives a way for testing and deriving desired objectives.
>
> > *“Table 1: it would be better to explain the level of the confidence interval (CI) in each entry, e.g., are they 95% CIs?”*
>
> In Table 1, we reported the average accuracy with standard deviation averaged over 5 random seeds. We have made it clear in the revised text.
>
>
> > *Typos & cross references*
>
> We thank the reviewer for carefully checking the text of our paper and finding the typos. We have fixed them in the revised paper.

---

> ### Author Response · Authors · 2021-11-18
> **Our theoretical framework can be potentially generalized with additional assumptions but those might not be realistic**
>
> > *”Would a more meaningful framework be to look at representation trained only on the source domain ... is there a way to generalize Theorem 1 and if so how much would the characterization change?”*
>
> Thank you for raising this! We think it is an interesting direction to generalize our theoretical framework and results to the setup where only partial target information is available. Our impossibility result implies that without access to target information we need to make additional assumptions. For example, one may seek to put additional assumptions between the source and target domains (e.g., the linear general position assumption in [4]). However, such assumptions are not easily satisfied or verified in practice, which is why we seek to achieve optimality with practical self-supervised learning.
>
> As a side note, one may also consider a practical semi-supervised domain adaptation scenario where we have access to source labeled data, a small number of labeled target data, and (possibly a lot) unlabeled target data. One could then possibly learn optimal representations by using the labeled target data to minimize target risk and unlabeled target data to enforce support match.
>
> [4] Arjovsky, Martin, et al. "Invariant risk minimization." arXiv preprint arXiv:1907.02893 (2019).

---

> ### Author Response · Authors · 2021-11-18
> **Our characterization of appropriate representations for DG provides guidance on finding augmentations in practice**
>
> > *”So there appears a gap in the theory: The theory requires a strict domain-covering augmenter, while in practice this can only be nearly satisfied”*
>
> We agree that strict domain-agnostic (DA) augmentation (we renamed “domain-covering” to “domain-agnostic” to avoid confusion) is hard to satisfy in practice.
>
> As you mentioned, the most important implication of our characterization of DA augmentations is to provide guidance on what kind of augmentations to use for learning robust representations in practice. Although we only prove the case for strict DA augmentations, we conjecture that more DA augmentations also lead to more robust representations. We empirically show that approximately DA augmentations is likely good enough in practice in Figure 3(c). This is also why we believe that CLIP (with its approximately DA image-text augmentations) gives much more robust representations than other SOTA SSL methods like DINO (standard augmentations) as seen in Table 1. We hope to extend our theory to soft domain-agnostic augmentations in future work.

---

> ### Author Response · Authors · 2021-11-18
> **Our gains are actually large for finetuning and could be larger for end-to-end training**
>
> > *“It seems that the benefit of domain bottlenecks is not very significant”*
>
> We believe that the improvements we find are more significant and promising than what the percentages show. As a comparison, [3] shows that none of recent DG methods could really outperform naive ERM using a unified benchmark DomainBed, which can be seen in our full results in Table 4 where the best algorithm (CORAL) only achieves about 0.6% improvement on average. Even the “DomainBed SOTA” in Table 1 which is the **best** result selected for each dataset **separately**, i.e., not achievable by any one method, only outperforms ERM by 0.5~2%. In contrast, our single method CLIP + CAD *consistently* outperforms CLIP + Base and raw CLIP by a similar margin.
>
>
> We would also like to emphasize that providing theoretical motivation (Sec 4.1) and empirical evidence (Table 1) for using CLIP as a robust DG model  is also a big part of our contributions. Our theoretical results further motivate the use of domain bottlenecks, and we empirically show that CLIP can be **consistently** improved even more with domain bottlenecks as shown in Table 1 & Table 2. We also believe that the improvement could be much larger if CLIP was trained end-to-end with our bottlenecks. As we do not have access to CLIP’s dataset, we unfortunately cannot train end-to-end CLIP with our bottleneck but hope that the above results will encourage others to do so.

---

> ### Author Response · Authors · 2021-11-18
> **We are the first to formalize optimal representations for DG, and our necessity result leads to minimal sufficient objectives**
>
> > *”I am not sure how much this distinguishes from the literature.” / ”how non-trivial is the necessity?” / ”Perhaps the main contribution here is to make this intuition concrete, or perhaps crafting the minimax optimality in IDG?”*
>
> First, we were happy to see that you felt that our characterization of optimal representations “provides guidance on designing a representation learning scheme under covariate shifts” and “is a potentially important result”. To clarify the novelty of these contributions:
>
> * To our knowledge, we are the first to study and formalize optimal representations for DG, which is one of our main theoretical contributions. As a result, even the characterization of sufficiency is novel, since previous results in DG only **hint** towards marginal matching objectives that could be sufficient for optimal DG (e.g. [1, 2]) using generalization bounds. These bounds can often be loose, and the implied sufficient conditions are neither necessary nor generally achievable.
> * Since our characterization is both sufficient and **necessary**, it describes the *exact* and *minimal* set of optimal IDG representations, which provides guidance for deriving the **least stringent** sufficient objectives for achieving them. Note that sufficient but not necessary conditions for optimal DG can be trivial and do not generally give insight into DG, e.g., in many cases the label Y is a representation that is optimal for DG.
> * Our necessity and impossibility results may explain why most practical DG methods fail to outperform the naive ERM [3]: they overlook the importance of target information.
> * We are the first to prove (and even suggest) that one can learn optimal representations with practical self-supervised learning (SSL). Furthermore, we prove the form of the SSL objectives and the properties of augmentations to achieve the desired representations.
>
> We included a more detailed comparison with previous work in the first part of the related work section and discussion in Sec 3.2.
>
> [1] Ben-David, Shai, et al. "Analysis of representations for domain adaptation." Advances in neural information processing systems 19 (2007): 137.
>
> [2] Ben-David, Shai, et al. "A theory of learning from different domains." Machine learning 79.1 (2010): 151-175.
>
> [3] Gulrajani, Ishaan, and David Lopez-Paz. "In Search of Lost Domain Generalization." International Conference on Learning Representations. 2020.

---

### Author Response · Authors · 2021-11-18
**We improved the paper based on the feedback of reviewers**

We thank all reviewers for their helpful feedback, it has helped us strengthen the paper.

We are glad that the reviewers found the problem that we address important and even fundamental for ML [hrZK, 1s9j], and the writing solid [1s9j]. Reviewers found our theory important, interesting, and stimulating [b5Jo, hrZK, 1s9j] and the empirical analysis informative [1s9j].

There were some concerns, which we address in our responses to individual reviewers. We also incorporated the feedback and uploaded the new version of our paper:
* We now better highlight our theoretical contributions [hrZK]: Specifically, our work is the first that considers optimal representations for DG and even the sufficient condition has only been hinted at in previous work. The entire formalism is new, while necessity has never even been suggested.
* We renamed the criteria of “domain-covering” augmentations to *“domain-agnostic”* augmentations, because we realized that it was being confused with support coverage.
* We further clarified the practicality of “domain-agnostic” augmentations [hrZK, b5Jo, 1s9j]: our characterization of “domain-agnostic” provides guidance on finding augmentations in practice, and many multi-modal augmentations would likely be domain-agnostic.
* We further clarified our contribution relative to CLIP [hrZK, b5Jo]: We highlighted that one of our important contributions is to give insights into the incredible robustness of CLIP in Sec 4.1, and clarified the significance of our results on DomainBed in Sec 6.2.

Please see individual responses below for other specific changes and more details.

---

### Decision · Program_Chairs · 2022-01-20

**Decision:**

Accept (Poster)

**Comment:**

The paper aims at characterizing conditions for optimal representations required for the domain generalization problem under covariate shift. Under the Idealized Domain Generalization (IDG), the paper provides a variational characterization of the optimal representation and shows a number of intriguing results: (i) optimal representation should remain discriminative  across domains, (ii)  the representation’s marginal support needs to be the same across source and target. (ii) It is also shown that without any target information, no representation can do uniformly well over constant representation, thus supporting the necessity of target knowledge. Finally, the paper provides practical objectives of the proposed variational characterization by self-supervised learning using data-augmentation with experimental results.

The reviewers had raised a number of concerns, many of which were alleviated through the responses provided by the authors. All in all, in the discussion period, the reviewers indicated the novelty of the results and their importance in learning good representations for the domain generalization problem. The reviewers still had reservations about the following points, and I strongly recommend to the authors to address these points in the revised version (please see the reviews for more details):

(i) For the experiments, it is clear that there is a noticeable gap between the derived theory and the experimental results. It was argued that alt-text augmentation of CLIP is one of the practical choices for (approximate) domain-agnostic augmentation, but it is difficult to verify this.

(ii) The empirical gain for the proposed method over CLIP is not very significant. As shown in Table 1 (or the complete results in Table 4), the performance of the proposed method is tightly related to the original CLIP method. In the case where the pre-trained representations of CLIP fail to cover the target set (see the TerraIncognita column in the table), the proposed method can be much worse than other alternatives. Thus I'm worried about the usefulness of the proposed method in practice.

(iii) One of the reviewers had asked about the importance of the necessity condition and its implications (e.g. in practice). Please, make sure to address this in the final version.

Also, there has been several recent works on learning disentangled representations for the domain generalization problem, using e.g. weakly-supervised approaches (Matsuura et al, '20), or model-based approaches (Robey et al, '21). It would be interesting to see how the results/ideas in this paper would connect to/improve those settings.